# FlashMoE: Fast Distributed MoE in a Single Kernel

**Osayamen Jonathan Aimuyo**
Cornell University
oja7@cornell.edu

**Byungsoo Oh**
Cornell University
bo239@cornell.edu

**Rachee Singh**
Cornell University
rs2293@cornell.edu

## Abstract

The computational sparsity of Mixture-of-Experts (MoE) models enables sub-linear growth in compute cost as model size increases, thus offering a scalable path to training massive neural networks. However, existing implementations suffer from low GPU utilization, significant latency overhead, and a fundamental inability to leverage task locality, primarily due to CPU-managed scheduling, host-initiated communication, and frequent kernel launches. To overcome these limitations, we develop FlashMoE, a fully GPU-resident MoE operator that fuses expert computation and inter-GPU communication into a single persistent GPU kernel. FlashMoE enables fine-grained pipelining of dispatch, compute, and combine phases, eliminating launch overheads and reducing idle gaps. Unlike existing work, FlashMoE obviates bulk-synchronous collectives for one-sided, device-initiated, inter-GPU (R)DMA transfers, thus unlocking payload efficiency, where we eliminate bloated or redundant network payloads in sparsely activated layers. When evaluated on an 8-H100 GPU node with MoE models having up to 128 experts and 16K token sequences, FlashMoE achieves up to **9**× higher GPU utilization, **6**× lower latency, **5.7**× higher throughput, and **4**× better overlap efficiency compared to state-of-the-art baselines—despite using FP32 while baselines use FP16. FlashMoE shows that principled GPU kernel-hardware co-design is key to unlocking the performance ceiling of large-scale distributed ML. We provide code at https://github.com/osayamenja/FlashMoE.

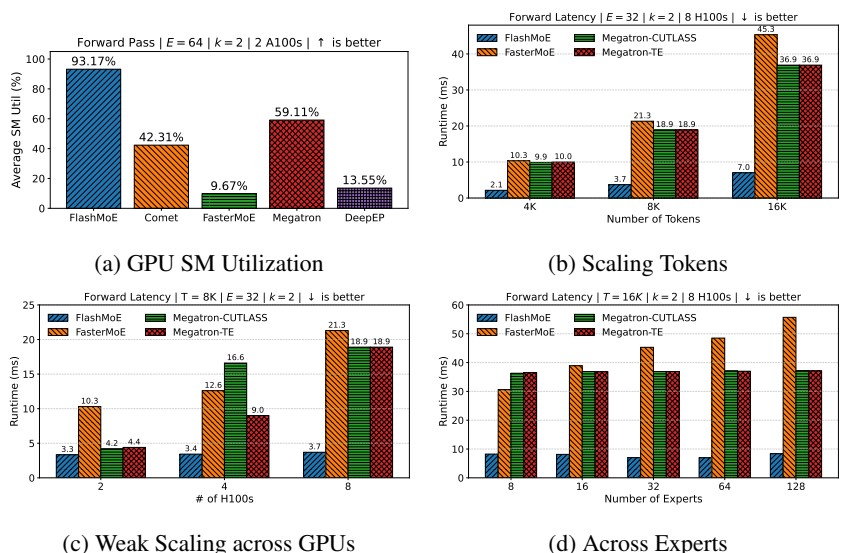

(a) GPU SM Utilization

(b) Scaling Tokens

(c) Weak Scaling across GPUs

(d) Across Experts

Figure 1: FlashMoE performance.

# 1 Introduction

State-of-the-art large language models (LLMs) [1–5] have adopted the Mixture-of-Experts (MoE) architecture for its computational efficiency and strong performance across a range of tasks. The traditional Transformer block consists of a self-attention module followed by a dense feed-forward network (FFN) [6]. In contrast, MoE architectures replace this single FFN (Figure 2(a)) with many identically sized FFNs, known as experts (Figure 2(b)). A trainable neural network, known as a gate function, sparsely activates these experts by dynamically routing input tokens to the experts selected at runtime. This increase in model parameters due to more FFNs improves model quality without the corresponding increase in computational cost.

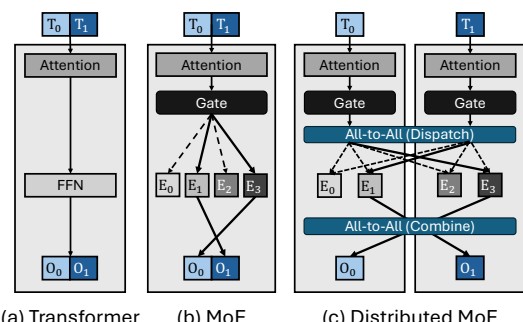

(a) Transformer    (b) MoE    (c) Distributed MoE

Figure 2: Transformer blocks (a) without MoE, (b) with MoE, and (c) with distributed MoE and expert parallelism. T, E, and O represent input tokens, experts, and output activations, respectively.

**Communication overheads in MoE.** As MoE model sizes grow, GPU memory constraints prevent hosting all experts on a single device. The standard practice is to distribute experts across multiple GPUs using expert parallelism (EP), which requires token routing via many-to-many communication primitives like *AlltoAll* [1, 4, 3, 7] (Figure 2(c)). Another round of *AlltoAll* communication restores the permuted tokens processed by experts to their original order in the sequence. *AlltoAll* communication is challenging to optimize on GPU networks and is highly sensitive to straggler delays — a phenomenon where a single straggler GPU delays all others from making progress [8]. These communication operations can account for 68% of the total runtime [9, 10], causing GPUs to remain idle (Figure 3, top left).

**Kernel launch overheads in MoE.** To mitigate these communication bottlenecks, recent work pipelines computation with communication kernels (Figure 3, left middle). However, the effectiveness of these solutions is limited by the overhead of launching many kernels from the CPU. Specifically, existing implementations [11–14] launch a large number of kernels per a single layer pass (see Table 1). Frequent kernel launches negatively affect performance by: (1) creating non-deterministic kernel start times across GPUs, exacerbating straggler issues; (2) introducing unnecessary synchronization points, causing GPUs to wait on peers or the CPU before proceeding; and (3) incurring repeated global memory round trips at kernel boundaries. Although CUDA graphs [15] can partially mitigate the first issue in static workloads, they are incompatible with MoE's dynamic expert routing patterns. Addressing the remaining issues requires novel solutions, which we provide in this work through complete kernel fusion and asynchronous device-initiated communication.

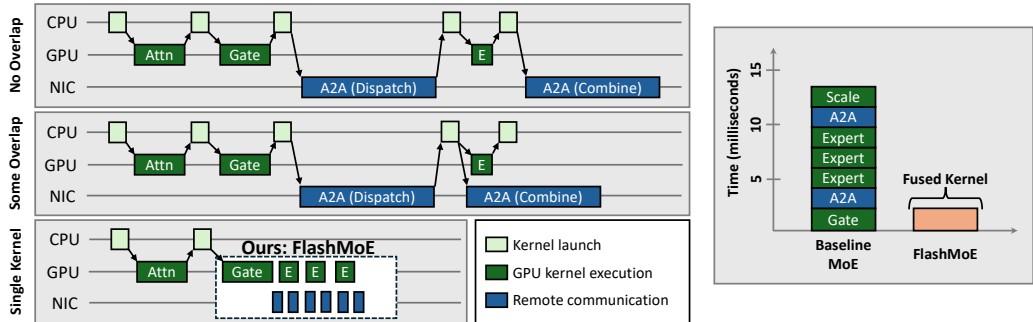

Figure 3: Comparing FlashMoE with state-of-the-art techniques that either do not overlap communication and computation (left, top) or do some overlap (left, middle). FlashMoE is a persistent kernel that fuses all computation and communication of the MoE operator (left, bottom). FlashMoE implements device-initiated computation (gate, expert FFN, scale) and communication tasks (right).

**Our Contributions: distributed MoE in a single kernel.** To overcome these fundamental inefficiencies in state-of-the-art MoE models, we develop FlashMoE a megakernel that integrates

all MoE computation and communication tasks into a single persistent GPU kernel *i.e.,* a kernel that remains active for the entirety of the MoE operator (Figure 3 bottom left). Instead of multiple kernel launches coordinated by the CPU, FlashMoE requires launching only one kernel, significantly reducing the involvement of the CPU. Within the fused kernel, FlashMoE implements a reactive programming model to achieve fine-grained parallelism and loosely coupled, non-blocking execution among tens of thousands of GPU threads.

**In-kernel Block scheduling and Tile parallelism.** FlashMoE implements *tile-level parallelism*, meaning it partitions input token matrices into smaller, independent units called *tiles*, which are processed by blocks but managed (scheduled and constructed) by warps. We specialize every thread block, except one, as *processors* to perform compute. In addition, we designate a dedicated Operating System (OS) block (4 warps) to perform administrative tasks of (1) scheduling computational work to processors (*scheduler*), and (2) decoding computational tasks from messages received from other GPUs (*subscriber*). This design allows FlashMoE to dynamically assign tasks to GPU blocks based on *readiness*, ensuring that no GPU SM remains idle throughout the lifetime of the MoE operator. FlashMoE selects tile dimensions to maximize GPU arithmetic intensity while benefitting from a high-degree of parallelism.

| MoE Implementation | GPU Ops |
|---|---|
| FlashMoE (this work) | 1 |
| COMET [12] | 33 |
| Megatron-LM CUTLASS [13, 16] | 85 |
| Megatron-LM TE [13, 16] | 261 |
| Megatron-LM + DeepEP [1] | 432 |
| DeepSpeedMoE [11] | 550 |

Table 1: We report number of GPU operations launched by MoE implementations by profiling with Nsight Systems [17]. We count operations in a single MoE layer (Gate → Dispatch → Expert → Combine) on 2 A100 GPUs, where each GPU has 32 experts. FlashMoE is the first to fully fuse the distributed MoE layer into a single GPU kernel.

**Asynchronous and payload-efficient communication.** By redesigning the MoE operator from the ground up, FlashMoE resolves fundamental inefficiencies inherent in the conventional MoE execution pipeline. One notable inefficiency is token padding during communication. To simplify programming complexity and due to symmetry constraints of collective communication APIs, existing implementations have to zero-pad token payloads to match predefined buffer sizes. This occurs when tokens are asymmetrically routed to experts, resulting in GPUs receiving much less than the expected capacity. However, these null payloads waste communication bandwidth, bloat data transfer latency and may lead to unnecessary computations on null matrices in some implementations. FlashMoE introduces *payload-efficient* communication by sending non-padded tokens only to GPUs with actively selected experts, conserving both communication and computational resources.

**Technical challenges.** Realizing the single-kernel design of FlashMoE required solving several technical challenges to achieve high performance: (1) lightweight computational dependency management; (2) navigating optimal SM occupancy configurations; (3) implementing in-device BLAS operations; (4) minimizing inter- and intra-device synchronization overheads; (5) implementing transfer-awareness by leveraging DMA over Unified Virtual Addressing (UVA) when available. In addressing these challenges, FlashMoE's design presents a radical departure from traditional synchronous *AlltoAll* collectives, where GPUs exhibit significant idle time during layer execution. For device-initiated communication, FlashMoE uses NVSHMEM [18] to establish a global address space across all GPUs for Direct Memory Access (DMA) communication. For in-device BLAS, FlashMoE develops custom high-performance GEMM operations via CUTLASS [19].

**Results.** Our evaluations show that FlashMoE achieves **6×** latency speedup, **9×** higher GPU utilization, **4×** better weak scaling efficiency and **5.7×** increased throughput compared to state-of-the-art implementations. We project these performance gains becoming even better in multi-node scenarios, where inter-node communication occurs using lower bandwidth inter-node links (*e.g.,* RDMA, Infiniband).

## 2 Motivation

**Synchronous Communication.** *AlltoAll* or *AllGather* communication as currently used in MoE frameworks is a *synchronous* collective operation, whose completion requires the participation of all involved GPUs. Here, disparities in processing speeds or kernel scheduling among GPUs induce

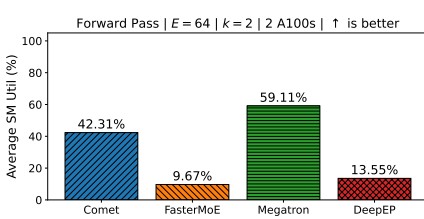
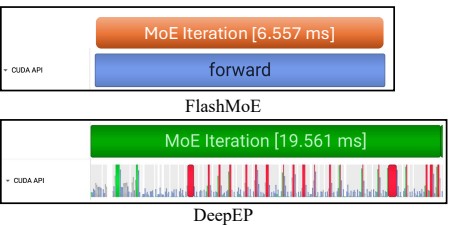

(a) GPU SM Utilization across baselines      (b) Kernel Launch overhead (CUDA API row)

Figure 4: 4a shows GPU utilization averaged across 100 MoE forward passes on 2 A100s with 300 GB/s unidirectional bandwidth, where we observe up to 90% idle time, due to kernel launch gaps and non-overlapping communication.

a straggler effect detrimental (Figure 13) to (1) the collective operation's performance and (2) E2E performance, as stalled GPUs cannot proceed to downstream dependent or independent tasks until the collective terminates. We expound on this problem in §A.

**Kernel Launch Overhead.** We compare the kernel launch overheads between FlashMoE and existing baselines. Table 1 shows the number of kernel launches during a single forward pass: FlashMoE launches exactly one persistent kernel, while the baselines launch up to 550 short-lived kernels to perform the same computation. Figure 4 provides a visual comparison using CUDA API traces captured by NSight Systems, illustrating the difference between FlashMoE and DeepEP. DeepEP exhibits many small CUDA API calls, with frequent stalls between individual operators, leading to increased GPU idle time (Figure 4a). In contrast, FlashMoE maintains high GPU utilization by avoiding launch overhead and synchronization gaps—achieving **93.17**% GPU utilization compared to 14% for DeepEP. See §4 for experimental results and §B for a discussion of related work.

## 3 Fused MoE Kernel Design

Modern distributed MoE systems suffer from two limitations: (1) frequent many-to-many (*AlltoAll or AllGather*) collectives on the critical path, and (2) significant overhead from repeated kernel launches. We address these in FlashMoE, a fully fused MoE operator implemented as a single persistent GPU kernel. Unlike previous approaches [12, 1, 11, 13, 20, 10, 21–25], FlashMoE is the first solution to implement a *completely fused Distributed MoE kernel*, eliminating kernel launch overhead entirely by requiring only a single kernel launch (see Table 1).

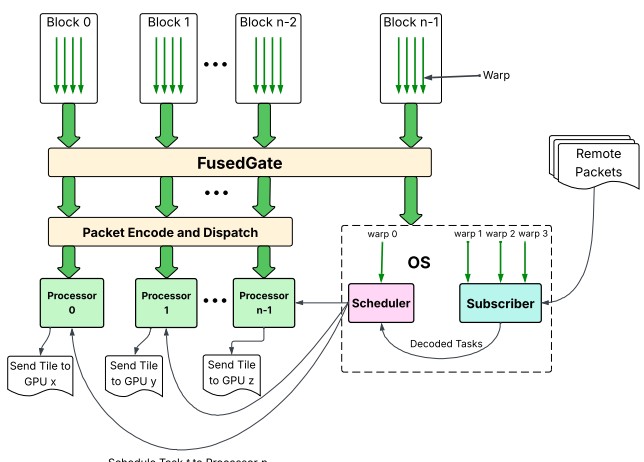

Figure 5: *FlashMoE Fused Kernel*

**Actor-based model.** The design of FlashMoE is based on the actor model of concurrent computation [26–28]. We implement this model by specializing GPU thread blocks and warps into three distinct actor roles: (1) **Processor** (§F.1), (2) **Subscriber** (§F.3), and (3) **Scheduler**(§F.2). The Processor performs compute (GEMMs and element-wise operations) and tile communication. We use CUTLASS [19] as the underlying infrastructure for high-performance BLAS routines and NVSHMEM for kernel-initiated communication [18]. The Subscriber and Scheduler perform administrative functions. Specifically, the Scheduler assigns computational tasks to available thread blocks. Our key innovation is making the Scheduler both *multithreaded*, enabling high scheduling throughput, and *work-conserving*, ensuring consistently high GPU SM utilization. On the other hand, the Subscriber decodes *tile packets* from peer GPUs to task

**Algorithm 1:** *FlashMoE Distributed MoE Fused Kernel*

---

**Input:** $A, O \in \mathbb{R}^{S \times H}$, $X \in \mathbb{R}^{E \times H \times D}$, $N$

1 **begin**
2      $T_\phi, G_\phi \leftarrow$ **FusedGate**$(A)$
3      **if** *blockId* $+ 1 < N$ **then**
4          **Dispatch**$(T_\phi, A)$
5          processor::start()
6      **else**
7          **if** $warpID == 0$ **then**
8              scheduler::start()
9          **else**
10              subscriber::start($T_\phi, G_\phi, O, X$)
11          **end if**
12      **end if**
13 **end**

---

$$D^j \xrightarrow{\text{Dispatch packets}} S_b^i \xrightarrow{\text{Notify Tasks}} S_h^i \xrightarrow[GEMM_0]{\text{Schedule Task}} P^i \xrightarrow{\text{Notify Tasks}} S_h^i \xrightarrow[GEMM_1]{\text{Schedule Task}} P^i \xrightarrow{\text{Send Tile}} S_b^j \xrightarrow{\text{Notify Tasks}} S_h^j \xrightarrow[\text{Combine}]{\text{Schedule Task}} P^j$$

Figure 6: *DMoE Functional Dependencies Expressed as a Chain of Actor Interactions*. We denote $S_b$, $S_h$, and $P$ as the Subscriber, Scheduler and Processor actors, respectively. For any actor $a \in \{S_b, S_h, P\}$, $a^i$ identifies an actor on GPU $i$. We define $D_i^j$ as the operator, where GPU $j$ dispatches packets of tiles to GPU $i$. This diagram expresses task dependencies at the granularity of a tile, namely $GEMM_0$, $GEMM_1$, combine and communication produce an output tile. Notifications occur as signals propagated through shared memory (subscriber $\leftrightarrow$ scheduler) or global memory (scheduler $\leftrightarrow$ processor or inter-GPU communication). Note one-sided inter-GPU transfers (packet or single tile) are *coupled* with a signal to notify $S_b^j$ on the receiving GPU $j$ of the message's delivery.

descriptors (§3.1). Of the $N$ thread blocks on a GPU, we specialize $N - 1$ to adopt the **Processor** role. We specialize the last block as the Operating System (OS). Within this block, we specialize three warps for the **Subscriber** role and one warp for the **Scheduler** role. This split of thread blocks across actors is intentional: our goal is to use few resources for administrative tasks while reserving bulk of the resources for performing MoE computation tasks. Figure 5 summarizes the FlashMoE architecture and its constituent actors, while Algorithm 1 gives a very close translation of the system in code. Note that $A \in \mathbb{R}^{S \times H}$ is the input token matrix; $O \in \mathbb{R}^{S \times H}$ the output matrix; and $X \in \mathbb{R}^{E \times H \times D}$ is a 3-D tensor of expert weights, where $E$ denotes the number of local experts for the executing GPU, $H$ is the embedding dimension, $D$ is the FFN intermediate dimension and $S$ is the sequence length. $T_\phi \in (\mathbb{N} \times \mathbb{R})^{E \times C}$ is a routing table data structure, where $T_\phi(e, c) = (i, w)$ indicates that token $i$ at slot $c$ dispatches to expert $e$. $w$ is the combine weight (Equation 2) and $C$ is expert capacity. The tuple structure of $T_\phi$ is an implementation detail. $G_\phi \in \mathbb{R}^{S \times E}$ captures the affinity scores produced by the gate (Equation 3). **Inter-actor interactions in FlashMoE.** FlashMoE decomposes MoE computation and communication at the granularity of a tile, a statically sized partition of a tensor, to achieve parallel execution and efficient overlap of tasks. Each tile maps to a discrete unit of work encapsulated by a *task descriptor*. The **Subscriber** decodes these task descriptors from the remote tile packets it receives. Concurrently, the **Scheduler** receives notifications about available tasks and dispatches them for execution to **Processor** actors that perform computations defined by these tasks, namely the feed-forward network (FFN) and expert-combine operations. Figure 6 show the chain of actor interactions, demonstrating how FlashMoE enforces DMoE functional dependencies.

**Determining tile dimensions in FlashMoE.** Selecting appropriate tile dimensions in FlashMoE is crucial to ensure efficient GPU utilization. An undersized tile underutilizes the GPU, while excessively large tiles create register pressure, causing performance-degrading register spills to local memory. After careful parameter sweeps, we choose tile dimensions of (128, 64). Our key insights are: increasing tile width significantly raises the register usage per thread, potentially triggering costly spills; increasing tile height without adjusting thread count increases workload per thread, harming performance. Raising the thread count per block beyond our fixed value of 128 threads reduces the

number of concurrent blocks, negatively affecting SM occupancy. Larger thread-block sizes also increase overhead from intra-block synchronization (__syncthreads() barriers), further degrading performance. Thus, our chosen tile dimensions balance register usage, shared-memory constraints, and GPU occupancy to deliver optimal performance.

## 3.1 Task Abstraction for Computation

**Computational operators.** The FFN operator is a standard position-wise feed-forward network widely used in Transformer architectures [6], composed of two linear transformations separated by a nonlinear activation $\phi$ (e.g., GELU or ReLU):

$$\text{FFN}(x) = W_2 \cdot \phi(xW_1 + b_1) + b_2 \tag{1}$$

Here, $W_1$ and $W_2$ represent learnable weight matrices, and $b_1$ and $b_2$ are biases. The expert-combine operation, used in architectures like GShard [29] and DeepSeek [1], merges outputs from multiple experts by computing a weighted combination based on their affinity scores:

$$\mathcal{C}_i = \sum_{j=1}^{k} g_{i,e} \tag{2}$$

$$\mathbf{h}_i = \sum_{j=1}^{k} \frac{g_{i,e}}{\mathcal{C}_i} \cdot \mathbf{h}_i^k \tag{3}$$

In these equations, $i \in 0, S-1$ represents an input token index, $e = E_{i,k}$ identifies the $k$-th expert selected for token $i$, and $g_{i,e}$ is the affinity score indicating how relevant expert $e$ is for token $i$.

**Unified task abstraction.** We unify the FFN and combine operations under a common abstraction called a *task*. Tasks provide a uniform interface for communicating tile-level work among Subscribers, Schedulers, and Processors. Formally, a task descriptor $t \in \mathcal{T}$ is defined as a tuple:

$$t = (\mathcal{M}, \star, \phi)$$

where $\mathcal{M}$ is a set of metadata (*e.g.,* device ID, tile index), $\star$ is a binary tensor operation (specifically, matrix multiplication $\cdot$ or Hadamard product $\odot$), and $\phi$ is an element-wise activation function (e.g., ReLU or identity).

We define a task $t$ operating on input tensors $A$, $B$, $D$, producing output tensor $C$, as follows:

$$\mathcal{F}_t(A, B, C, D) \coloneqq C \leftarrow \phi(A \star_t B + D) \tag{4}$$

The operator $\star_t$ (instantiated from $\star$) may behave differently depending on the task metadata $\mathcal{M}$, and the result of $A \star_t B$ is accumulated into $D$. We provide an example of task metadata in §E.

In practice, we implement each task defined by Equation 4 as a *single fused* __device__ decorated function which the **Processor** (Algorithm 2) invokes at runtime. Fusion for $t$ entails applying $\phi$ and the succeeding addition operation to registers storing the results of the binary operator $\star_t$. To illustrate its flexibility, we show how the FFN and expert-combine operations can be expressed using this task framework. Note that we omit the matrix multiplication symbol ($\cdot$) for simplicity. Also, $\phi_1$ can be any activation function, while $\phi_2$ is the identity function. The FFN is expressed as:

$$t_1 = (\mathcal{M}, \cdot, \phi_1), \quad t_2 = (\mathcal{M}, \cdot, \phi_2),$$
$$\mathcal{F}_{t_1}(A, B_1, C_1, D_1) \coloneqq C_1 \leftarrow \phi_1(AB_1 + D_1),$$
$$\mathcal{F}_{t_2}(C_1, B_2, C_2, D_2) \coloneqq C_2 \leftarrow \phi_2(C_1 B_2 + D_2).$$

Whereas, the expert-combine operation is formalized as:

$$t_3 = (\mathcal{M}, \odot, \phi_2),$$
$$\mathcal{F}_{t_3}(A, S, C, C) \coloneqq C \leftarrow \phi_2(A \odot S + C).$$

## 3.2 Symmetric Tensor Layout for Inter-GPU Communication

Within a single GPU device, the actors in FlashMoE communicate through the GPU's memory subsystem. Specifically, the Scheduler and Subscriber actors exchange data via fast shared memory,

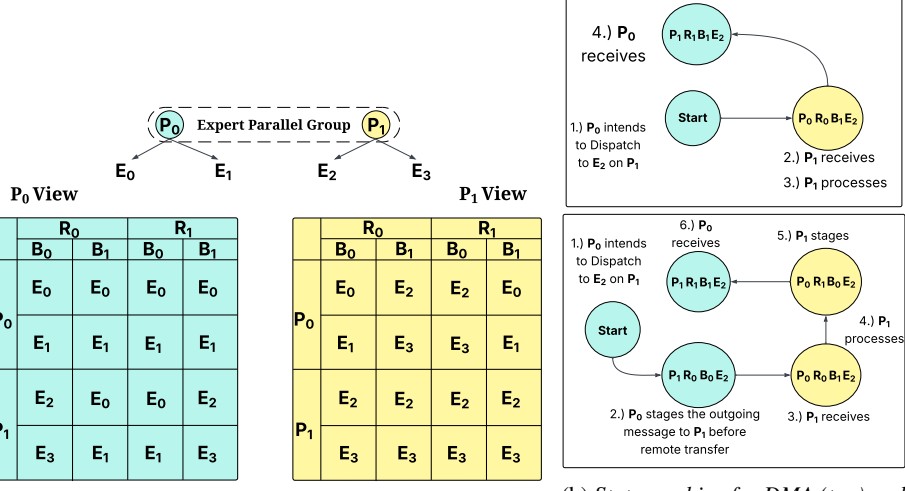

(a) *Layout across 2 Expert-parallel Processes.*

(b) *State machine for DMA (top) and RDMA (bottom) communication.*

Figure 7: Symmetric Tensor Layout

while other actor pairs communicate through global memory. For communication across multiple devices, FlashMoE uses *device-initiated communication*, leveraging the one-sided PGAS (Partitioned Global Address Space) programming model [30]. However, achieving scalable and correct one-sided memory accesses in PGAS without costly synchronization is a known challenge [1, 31]. We address this challenge with a provably correct and scalable solution: a symmetric tensor layout $L$, supporting fully non-blocking memory accesses. We define L as:

$$L \in \mathbb{R}^{P \times R \times B \times E \times C \times H}$$

where: $P$ is the expert parallel world size, $R$ identifies communication rounds (*i.e.,* two rounds, one for token dispatch and one for combine), $B$ is number of staging buffers, $E$ is the number of local experts, $C$ is the upscaled expert capacity (§3.2.1) and $H$ is the token embedding dimension. Our core insight to enable non-blocking communication is *temporal buffering*. Specifically, we overprovision memory for the underlying token matrix by at least $2 \cdot r$ times, where $r$ is the number of communication rounds in the dependency graph, and the factor of 2 accounts for separate buffers for incoming and outgoing data within each communication round. For MoE models, we have $2 \cdot r = 4$. This modest increase in memory usage eliminates the need for synchronization during one-sided data transfers. Figure 7b illustrates how cells within this symmetric tensor layout are indexed and used for Direct Memory Access (DMA) and Remote DMA (RDMA) operations. As Theorem 3.1 reinforces, this indexing scheme over $L$ is the underlying mechanism that allows for fully non-blocking accesses eliding synchronization because all accesses are write *conflict-free*. See§ C for the proof.

**Theorem 3.1.** *The symmetric tensor layout $L$ is write-write conflict-free.*

To construct $L$, we start from the original token buffer $T \in \mathbb{R}^{S \times H}$, where $S$ is the sequence length and $H$ is the token embedding dimension. We first reorganize the sequence dimension $S$ into three sub-dimensions representing the expert capacity ($C$), local expert slots ($E$), and the expert parallel world size ($W$), st:

$$C \cdot E \cdot W = C \cdot E' = S', \quad \text{where} \quad S' \geq S \text{ and } E' \geq E_W$$

In the typical case of uniform expert distribution (illustrated in Figure 7a), we have $S' = S$ and $E' = E_W$, where $E_W$ is the total number of experts in the model. Thus, the size of the token buffer is $Size(T) = S' \cdot H$. In Figure 7a, each cell labeled $E_i$ (with $i \in \{0, \ldots, 3\}$) is a matrix of size $(C, H)$. Extending prior work [29, 12], we introduce additional temporal dimensions $R$ (communication rounds) and $B$ (staging buffers). Each communication round has two fixed staging slots: one for outgoing tokens and another for incoming tokens. Each slot, indexed by dimension $P$, forms a tensor of shape $(S', H)$. Therefore, the tensor size $Size(L)$ is generally at least four times the original token buffer size, becoming exactly four times larger in the case of uniform expert distribution. Empirically, we find $Size(L) \approx 4 \cdot Size(T)$, contributing memory overhead $\leq 2\%$ of memory capacity for inference of popular models. We present a thorough breakdown in §D.

### 3.2.1 In-place Padding for Payload Efficiency

Due to the dynamic and uneven distribution of tokens in MoE dispatch [32], GPUs commonly receive fewer tokens than their predefined expert capacity. Current MoE frameworks [11] typically pad these buffers with null tokens before computation, unnecessarily increasing communication payloads and degrading performance. In contrast, we propose *in-place padding*, performing padding directly within the local symmetric tensor buffers and thus eliminating excess network communication.

As we show in Figure 7a as a reference, each cell $E_i$ is sized according to the expert capacity $C$. We further align this capacity to ensure divisibility by the tile block size $bM = 128$, guaranteeing safe and aligned memory reads by Processor threads consuming remote tokens. This in-place padding strategy slightly increases the memory footprint of $L$, as described below:

$$Size(L) \approx \begin{cases} 4 \cdot Size(T), & \frac{S}{E} \geq bM \\ 4 \cdot \frac{bM \cdot E}{S} \cdot Size(T), & \text{otherwise} \end{cases}$$

## 4 Evaluation

We implement (§G) and evaluate FlashMoE across five metrics: **Forward Latency** (§ 4.1), **GPU Utilization** (§ 4.2), **Overlap Efficiency** (§ 4.4), **Throughput** (§ 4.3), and **Expert Scalability** (§ 4.5). We run experiments on a server with 8 NVIDIA H100 80G GPUs interconnected via NVLink, 125 GB of RAM, and 20 vCPUs. We used PyTorch 2.6.0, CUDA 12.8, and Ubuntu 22.04. All experiments use MoE transformer models configured with 16 attention heads, an embedding dimension of 2048, and an FFN intermediate size of 2048. We apply Distributed Data Parallelism (DDP) and Expert Parallelism for all experiments. We execute only the forward pass over a single MoE layer and measure the average runtime of 32 passes after 32 warmup passes. We use top-2 routing with a capacity factor of 1.0. We compare FlashMoE against several state-of-the-art MoE systems: (1) **Comet** [12], (2) **FasterMoE** [14], (3) **Megatron-CUTLASS** [13], and (4) **Megatron-TE**: Megatron-LM with Transformer Engine [33]. Comet relies on `cudaMemcpyPeerAsync` [34], while FasterMoE and Megatron-LM use NCCL exclusively for communication.

**Desiderata.** We observe Comet exhibiting anomalously bad performance values at 8 GPUs, so we exclude their results from evaluations at 8 GPUs and only include for results at ≤ 4 GPUs. We evaluate FlashMoE using FP32 precision whereas all baselines use FP16. We do so because (1) of incomplete fp16 tuning (§H) and (2) no baseline supports FP32. Note, this precision discrepancy disadvantages FlashMoE by doubling its communication volume and computational workload.

### 4.1 Forward Latency

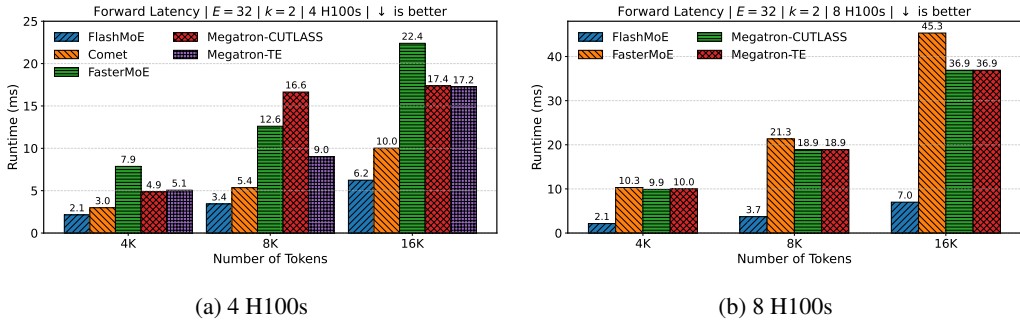

(a) 4 H100s                    (b) 8 H100s

Figure 8: Forward Latency as the *Number of Tokens* per GPU increases.

We first measure the forward latency of FlashMoE across different sequence lengths on both 4 and 8 GPU setups (Figure 8). FlashMoE consistently outperforms all baselines, with especially notable improvements at longer sequence lengths. On 4 GPUs, it achieves up to **4.6**x speedup over Megatron-TE at 16K tokens, and **2.6**x over FasterMoE. The gains are even more pronounced at 8 GPUs where FlashMoE maintains low latency, exhibiting up to **6.4**x speedup over baselines that degrade steeply due to increasing communication costs as token buffers increase proportionally.

## 4.2 GPU Utilization

To quantify GPU efficiency, we measure Streaming Multiprocessor (SM) utilization during the forward pass (Figure 9). FlashMoE achieves 93.17% average SM utilization, over **9**x higher than FasterMoE (9.67%), **6.8**x higher than DeepEP+Megatron-LM (13.55%) **4**x higher than Megatron-TE (59.11%), and **2.2**x higher than Comet (42.31%). This improvement stems from our fully fused kernel architecture and fine-grained pipelining of compute and communication tasks. By eliminating idle gaps due to kernel launches and enabling in-kernel task scheduling, FlashMoE ensures SMs remain busy with productive work throughout execution.

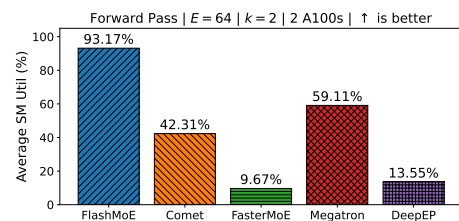

Figure 9: SM utilization, defined as the ratio of cycles in which SMs have at least one warp in flight to the total number of cycles [17]. Values represent the average SM utilization over 100 iterations.

## 4.3 Throughput

As shown in Figure 10, FlashMoE scales linearly with GPU count, reaching 17.7 MTokens/s at 8 GPUs. This is over **5.7**x higher than FasterMoE and **4.9**x higher than Megatron-TE and Megatron-CUTLASS. Notably, these results are achieved despite *FlashMoE operating entirely in FP32, while baselines use FP16*. This indicates that FlashMoE 's design eliminates throughput bottlenecks not by exploiting lower precision, but by maximizing hardware utilization and eliminating host-driven inefficiencies.

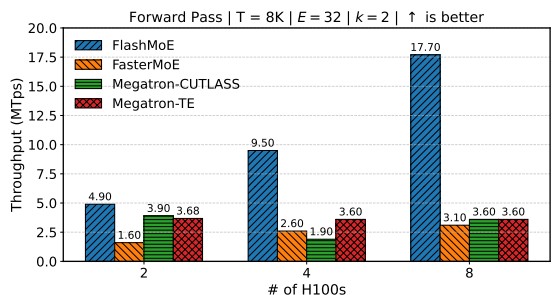

Figure 10: Throughput when scaling the number of GPUs, computed as $\frac{T \times N_G}{\text{latency}}$.

## 4.4 Overlap Efficiency

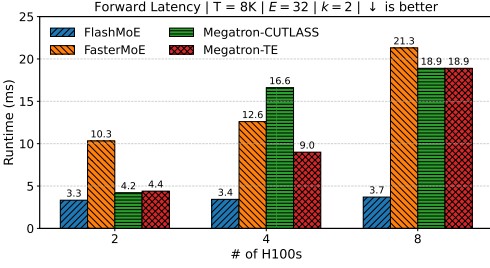

(a) Latency as Number of GPUs increases.

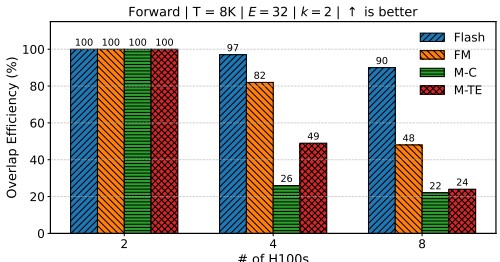

(b) Weak scaling efficiency

Figure 11: Weak scaling efficiency. We define Overlap Efficiency $O_e$ to be $O_e = T(2)/T(N_G)$, where $T(N_G)$ is the latency at $N_G$ GPUs and $T(2)$ is the latency at 2 GPUs.

We evaluate the extent to which FlashMoE overlaps communication and computation by measuring weak scaling efficiency as the number of GPUs increases (Figure 11b). We note that most baselines fail to execute at a single GPU, hence why we use 2 GPUs as the reference point. We observe that Megatron-CUTLASS and Megatron-TE degrade significantly, with overlap efficiency dropping below 50% at $\geq 4$ GPUs. FlashMoE gives up to **3.88**x and **4**x higher efficiency at 4 and 8 GPUs, respectively. Figure 11a further illuminates this efficiency, as FlashMoE shows stable forward latency growth. These results corroborate that FlashMoE's actor-based design and asynchronous data movement achieve near-ideal overlap.

## 4.5 Expert Scalability

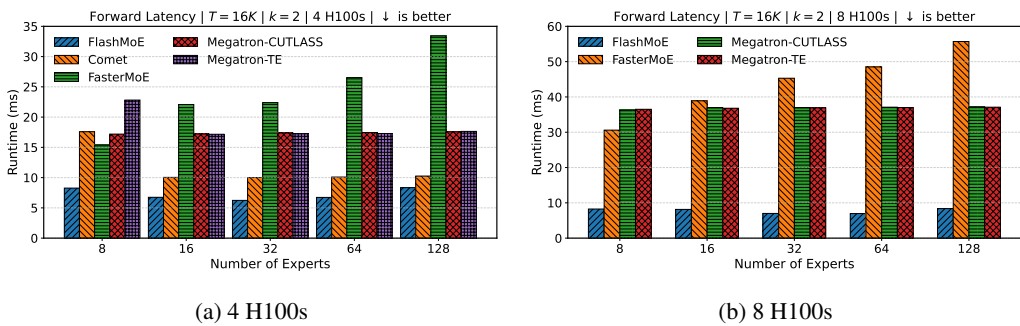

(a) 4 H100s

(b) 8 H100s

Figure 12: Forward Latency as the *Number of experts* increases.

We analyze how FlashMoE scales with increasing number of experts at fixed sequence length (T = 16K). Note that for the discussed plots, the number of experts on the x-axis is the *total number across all GPUs*. Each GPU gets 1/8th of this value. As seen in Figure 12, FlashMoE maintains *low, uniform* latency, as desired, even as the number of experts grows from 8 to 128. In contrast, baselines exhibit superlinear latency increases due to increased kernel launch overheads. FlashMoE outperforms these baselines by up to **4**X at 4 H100s and **6.6**X at 8 H100s, both at 128 experts. FlashMoE 's payload-efficient communication and scheduler-driven in-kernel dispatching allow it to sustain expert parallelism without incurring the communication and orchestration penalties seen in other systems. These results reinforce FlashMoE 's scalability for ultra-sparse MoE configurations.

## 5 Limitations and Future Work

**Engineering complexity.** Fully fused, persistent kernels demand deep GPU + distributed-systems expertise; future work may investigate compiler/DSL abstractions to lower this barrier.

**FP16 inefficiency.** Our FP16 path is suboptimal (§H) due to insufficient tuning. We anticipate addressing this gap with autotuned GEMM operators like cuBLASDx [35] or CUTLASS builders.

**Training support.** This work targets inference; enabling training will require fusing backward computation and gradient communication with new bookkeeping and task descriptors.

## 6 Conclusion

We introduce FlashMoE, the first work to fuse the entire Distributed MoE operator into a single persistent GPU kernel that unifies computation, communication, and scheduling via actor-style concurrency, warp specialization, and async (R)DMA. We address two dominant bottlenecks in prior systems—CPU-managed synchronous communication and fragmented multi-kernel execution. Empirically, FlashMoE achieves up to **6×** speedup, **9×** higher GPU utilization, and **5.7×** throughput for distributed MoE. Looking ahead, we see a shift from CPU orchestration to fully autonomous, GPU-native pipelines—extending this fusion approach to training and beyond.

## 7 Acknowledgements

This research is supported by NSF Award #2444537 and ACE, one of the seven centers in JUMP 2.0, a Semiconductor Research Corporation (SRC) program sponsored by DARPA. This work also used resources of the National Energy Research Scientific Computing Center, a DOE Office of Science User Facility supported by the Office of Science of the U.S. Department of Energy under Contract No. DE-AC02-05CH11231 using NERSC award ASCR-ERCAP0030076. We acknowledge and thank Dr. Giulia Guidi for providing access to these NERSC supercomputing resources.

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

# A    Supplementary Motivation

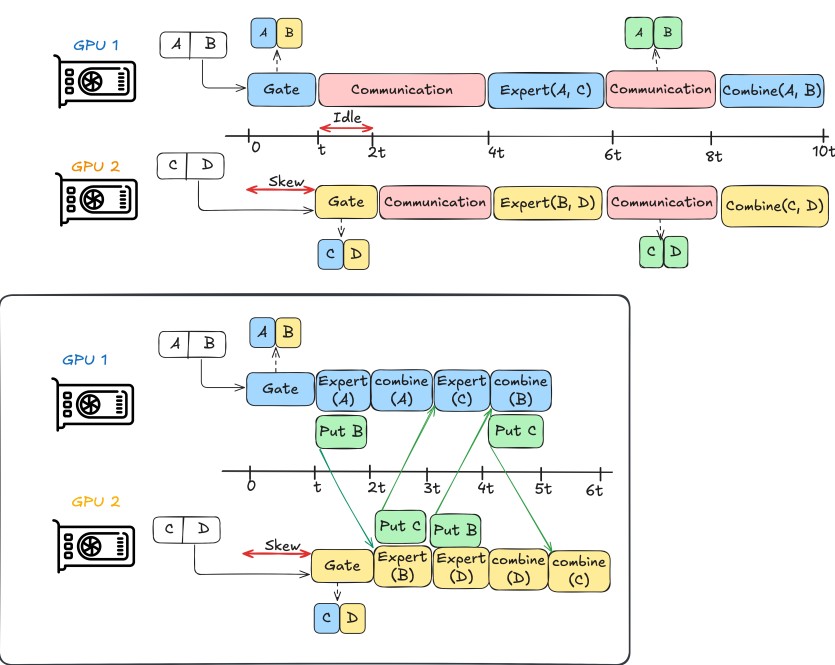

Figure 13: Overlapped Schedule (bottom) showing how idle time from the sequential schedule (top) is repurposed for computation. FlashMoE implements the overlapped schedule.

In Figure 14, we present empirical cumulative and raw distributions of *AlltoAll* kernel runtime from distributed training of a 1.3B GPT-3 MoE model across 32 A100 and 8 V100 GPUs. We use this result to motivate the severity and prevalence of straggler effects. In Figure 14b, we observe P95 communication performance degradation of **1.32X** when compared to the mean actual kernel time. This performance reduction is rather tame as the underlying hardware is a supercomputer well-tuned against "software jitter" [36]. However, we observe a more severe p95 performance loss of **11X** in a single-node Virtual Machine (VM). In line with prior HPC works [37, 38], we argue that obviating the inherent barrier in this synchronous collective communication would allow GPUs to repurpose this observed idle time for downstream computation as depicted in Figure 13.

Table 2: Straggler Delay within Synchronous *All-to-All* communication. We capture the distribution of delay induced by stragglers across many steps. Let **Actual Time** $t_a$ denote the fastest kernel execution time across all GPUs, and **Total Time** $t$ be the maximum recorded step time. We define $Delay$ as the maximum difference between $t$ and $t_a$. Note $Delay$ is idle time. For the 1x8 V100, we profile 1750 steps and 600 steps for the 8x4 A100. See Figure 14 for the raw distribution.

| System | # Nodes | # GPUs | Median | p95 |
|---|---|---|---|---|
| Commercial VM (V100) | 1 | 8 | 3.1x | 11.4x |
| Supercomputer (A100) | 8 | 32 | 1.09x | 1.32x |

# B    Related Work

**Computation-Communication Overlap and Kernel Fusion.** To reduce the communication overheads of synchronization in distributed DNN training, many research efforts have been focused on increasing the overlap of computation and communication. For generic Transformer-based models without MoE layers, many works [39–47] have provided insights and techniques to partition and schedule computation and communication operations, aimed at finer-grained overlapping. To address the challenges posed by *AlltoAll* communication and expert parallelism in MoE training, Tutel [48]

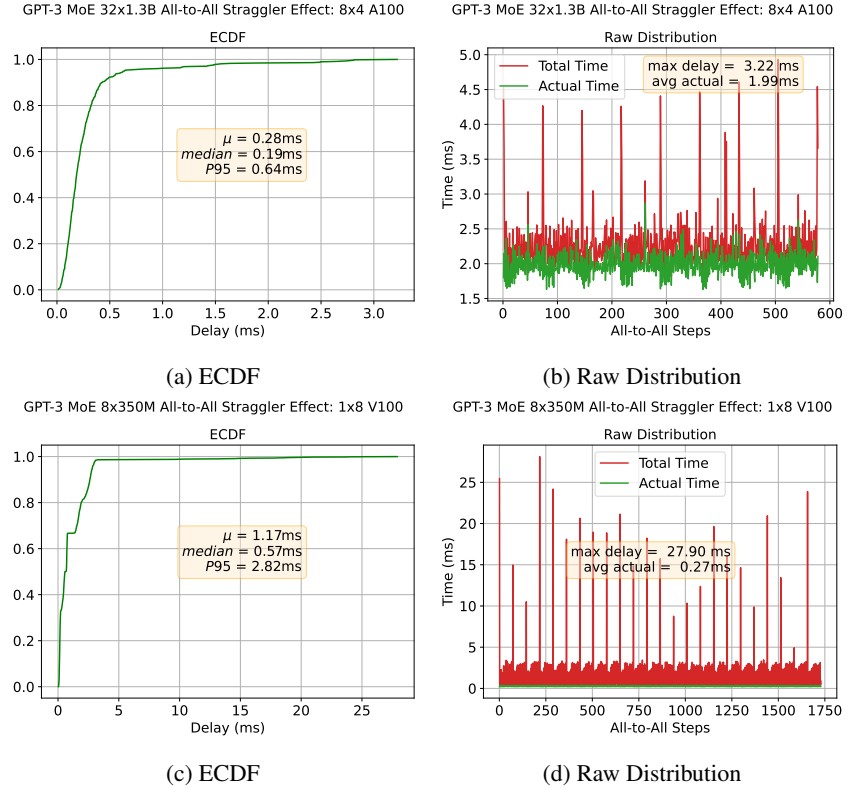

Figure 14: Straggler effect of synchronous *AlltoAll*. $M \times N$ A100 or V100 denotes $N$ GPUs within a node across $M$ nodes. Every GPU communicates with every other GPU per *AlltoAll* step. We capture the distribution of delay induced by stragglers across many steps. **Actual Time** $t_a$ denotes the fastest kernel execution time across all GPUs, conversely **Total Time** $t$ is the maximum recorded step time, while $Delay$ is the maximum difference between $t$ and $t_a$. Note $Delay$ is idle time.

and FasterMoE [14] overlap *AlltoAll* with expert computation. Lancet [49] additionally enables both non-MoE computation in forward pass and weight gradient computation in backward pass to be overlapped with *AlltoAll*. Despite overlapping, the performance of these approaches is limited in practice due to blocking synchronous collective communication with barriers. In contrast, Flash-MoE fundamentally eliminates these inefficiencies with asynchronous, device-initiated data transfers overlapped with tiled computation all *within a single kernel*. FlashMoE further differentiates itself from SOTA works like COMET [12] and DeepEP [1], which also use this form of kernel-initiated communication but at a coarse-grained granularity and without complete kernel fusion.

## C  Proof of Theorem 3.1

We begin with two necessary definitions vital to the proof.

**Definition C.1.** *Define a write as $w(p_s, p_t, i)$, where $p_s$ is the source process and $i$ is an ordered tuple indicating the index coordinates for $L$ residing on the target process $p_t$. A write-write conflict occurs when there exist at least two distinct, un-synchronized, concurrent writes $w_1(p_{s_1}, p_{t_1}, i_1)$ and $w_2(p_{s_2}, p_{t_2}, i_2)$, such that $p_{t_1} = p_{t_2}$ and index coordinates $i_1 = i_2$ but $p_{s_1} \neq p_{s_2}$*

**Definition C.2.** *For any source process $p_s$, a valid index coordinate $i = (p*, r, b, e, c)$ satisfies the following:*

1. *For inter-device writes, it must hold that $p* = p_s$ and $b = 1$. Note this also applies to self-looping writes $w(p_t, p_t, i)$.*

2. *For any write $w(p_s, p_t, i)$, if $b = 0$, then $p_s = p_t$. This rule describes intra-device staging writes.*

We restate Theorem 3.1 and outline its proof below.

**Theorem C.1.** *The symmetric tensor layout $L$ is write-write conflict-free.*

*Proof.* As is the case for typical physical implementations, assume that each index coordinate $i$ maps to a distinct memory segment in $L$. Next, we show by contradiction that no write-write conflicts can exist when accessing $L$ using *valid* $i$. For simplicity, we only include the index coordinates when describing a write. Assume that there exist at least two writes $w_1(p_{s_1}, p_{t_1}, i_1)$, $w_2(p_{s_2}, p_{t_2}, i_2)$ with $p_{t_1} = p_{t_2}$ and valid destination coordinates $i_1, i_2$, where $i_1 = i_2$ lexicographically and both are unpacked below.

$$i_1 = (p_1, r_1, b_1, e_1, c_1), \ i_2 = (p_2, r_2, b_2, e_2, c_2)$$

Note that intra-process writes always have distinct $c_j$ coordinates, where $j \in \{0, C - 1\}$. For inter-process transfers, we have two cases.

*Case 1: $p_{s_1} = p_{s_2}$*
Here, $w_1$ and $w_2$ are identical operations. This contradicts the definition of a conflict, which requires that $p_{s_1} \neq p_{s_2}$. In practice, such repeat writes never even occur.

*Case 2: $p_{s_1} \neq p_{s_2}$*
To ensure validity for $i_1$ and $i_2$, it is the case that $p_1 = p_{s_1}$ and $p_2 = p_{s_2}$. However, this implies that $i_1 \neq i_2$ yielding a contradiction as desired. $\square$

# D   Memory Overhead

We measure the GPU memory required for the symmetric tensor $L$ and runtime bookkeeping state of FlashMoE. The memory overhead primarily depends on the tile size, expert capacity ($EC$), and the number of experts ($E$). Table 3 summarizes the memory overhead across recent MoE models [50–55] during inference, showing that FlashMoE maintains a modest and predictable memory footprint. In particular, the symmetric tensor ($ST$) accounts for at most 2.15% additional memory relative to the total inference memory requirements.

Table 3: Memory overhead of FlashMoE (tile size $bM = 128$, $Size(T) = \text{Tokens} \times 4\text{KB}$).

| Model | Params | S | E | H | I | ST (GB) | Model (GB) | Overhead (%) |
|---|---|---|---|---|---|---|---|---|
| Moonlight-16B-A3B | 16B | 8K | 64 | 2K | 1.38K | 0.25 | 59 | **0.49** |
| Grok-1 | 314B | 8K | 8 | 6K | 32K | 0.75 | 1169 | **0.15** |
| Snowflake-Arctic | 479B | 4K | 128 | 7K | 4.75K | 1.75 | 1784 | **0.12** |
| Qwen3-235B-A22B | 235B | 40K | 128 | 4K | 1.5K | 3.00 | 875 | **0.38** |
| Mixtral 8x7B | 47B | 32K | 8 | 4K | 14K | 2.00 | 175 | **2.15** |
| DeepSeek-V3 | 685B | 160K | 256 | 7K | 2K | 1.50 | 2551 | **0.11** |

## E   Task Implementation

```
 1  #define GEMMs 2
 2  struct __align__(16) Task {
 3      const byte* aData;
 4      array<const byte*, GEMMs> bData;
 5      array<byte*, GEMMs> cData;
 6      array<const byte*, GEMMs> dData;
 7      byte* rcData;
 8      uint64_t* flags;
 9      uint M;
10      uint syncIdx;
11      uint tileIdx;
12      uint batchIdx;
13      uint peerIdx;
14      uint expertIdx;
15      uint isPeerRemote;
16      TaskType taskType;
17      uint16_t tileSize;
18      // Pad till 128-byte cache line
19      uint padding[6] = {};
20  }
```

Figure 15: *Task Struct*. TaskType $\in \{GEMM_0, GEMM_1, Combine\}$

## F  Actors

### F.1  Processor

---

**Algorithm 2:** *Processor Actor*: executed by a block

---

**1 begin**
**2**    $tQ \leftarrow$ **GetTQ**()
**3**    $signal \leftarrow 0$
**4**    // shared memory variables
**5**    $task \leftarrow \{\}$
**6**    $interrupt \leftarrow$ **False**
**7**    $complete \leftarrow$ **False**
**8**    **while** $interrupt ==$ **False do**
**9**      **if** $warpId == 0$ **then**
**10**        **if** $threadId == 0$ **then**
**11**          **awaitTaskFromScheduler**($interrupt$, $signal$)
**12**          **FencedNotifyRQ**($ready$)
**13**        **end if**
**14**      **syncwarp**()
**15**      **warpReadTQ**($tQ$, $signal$, $task$)
**16**    **end if**
**17**    **syncthreads**()
**18**    **if** $interrupt ==$ **False then**
**19**      **switch** *task.Type* **do**
**20**        **case** $GEMM_0$ **do**
**21**          // fused GEMM, epilogue and async tile staging
**22**          **fGET**($GEMM_0$, $task$)
**23**          **if** $threadId == 0$ **then**
**24**            $complete \leftarrow$ **NotifyTileCompletion**()
**25**          **end if**
**26**          **syncthreads**()
**27**          **if** $complete ==$ **True then**
**28**            **NotifySchedulerNextGEMM**($tQ$)
**29**          **end if**
**30**        **end case**
**31**        **case** $GEMM_1$ **do**
**32**          // fused GEMM, epilogue and async tile transfer
**33**          **fGET**($GEMM_1$, $task$)
**34**        **end case**
**35**        **case** $Combine$ **do**
**36**          **combine**($task$)
**37**        **end case**
**38**      **end switch**
**39**    **end if**
**40**    **end while**
**41 end**

---

### F.2 Scheduler

---

**Algorithm 3:** *Scheduler Actor*: executed by one warp

---

**1 begin**
**2**     $scheduled \leftarrow 0$
**3**     $tTB \leftarrow 0$
**4**     $tqState \leftarrow \{\}$
**5**     $pTDB \leftarrow$ **GetProcessorDoorbell()**
**6**     $sTDB \leftarrow$ **GetSubscriberDoorbell()**
**7**     $taskBound \leftarrow$ **GetTaskBound()**
**8**     $tTB \leftarrow$ **AtomicLoad**$(taskBound)$
**9**     // circular buffer ready queue
**10**     $rQ \leftarrow \{\}$
**11**     // Populate ready queue with Processor ids
**12**     **PopulateRQ**$(rQ)$
**13**     **while** $scheduled < tTB$ **do**
**14**        $lt \leftarrow 0$
**15**        **do in parallel**
**16**           Sweep doorbells and populate observed task counts into $tqState$
**17**           Aggregate locally observed task counts into $lt$
**18**        **end**
**19**        $qS, taskTally \leftarrow 0$
**20**        // qS is the inclusive output
**21**        **WarpInclusiveSum**$(lt, qS, tasktally)$
**22**        **while** $tasktally > 0$ **do**
**23**           Repopulate $rQ$ with ready processor ids
**24**           **do in parallel**
**25**              Starting at $rQ[qS]$, signal processors about task indices from $tqState$
**26**           **end**
**27**        **end while**
**28**        **if** $threadId == 0$ **then**
**29**           $tTB \leftarrow$ **AtomicLoad**$(taskBound)$
**30**        **end if**
**31**        $tTB \leftarrow$ **WarpBroadcast**$(tTB)$
**32**     **end while**
**33**     **InterruptSubscribers()**
**34**     **InterruptProcessors()**
**35 end**

---

### F.3 Subscriber

---

**Algorithm 4:** *Subscriber Actor*: executed by three warps

---

**Input:** $T_\phi \in \left(\mathbb{R}^2\right)^{E \times C}, G_\phi \in \mathbb{R}^{S \times E}, O \in \mathbb{R}^{S \times H}, X \in \mathbb{R}^{E \times H \times D}$

**1 begin**

**2**     $interrupt \leftarrow$ **GetSharedInterrupt**()

**3**     $flags \leftarrow$ **GetSymmetricFlags**()

**4**     $tQ \leftarrow$ **GetTQ**()

**5**     // Predefined upper bound on the number of tasks.

**6**     // We modulate this value to the actual task count computed

**7**     // dispatch signals received from peer GPUs

**8**     $taskBound \leftarrow$ **GetTaskBound**()

**9**     **while AtomicLoad**($interrupt$) == **False do**

**10**        // dispatch flags

**11**        **do in parallel**

**12**           Visit dispatch flags

**13**           Atomically retrieve signal

**14**           **if** *Signal is set and flag is not visited* **then**

**15**              Mark visited

**16**              **SelfCorrectTaskBound**($taskBound, Signal$)

**17**              Enforce memory consistency before consuming packet

**18**              Decode packet into a set of $GEMM_0$ task descriptors using $X$

**19**              Write task descriptors to $tQ$

**20**              Notify Scheduler of decoded tasks

**21**           **end if**

**22**        **end**

**23**        Advance flags by number of dispatch flags length

**24**        Atomically retrieve signal

**25**        // combine signals

**26**        **do in parallel**

**27**           Visit combine flags: one per tile

**28**           **if** *Signal is set and flag is not visited* **then**

**29**              Mark visited

**30**              Enforce memory consistency before consuming packet

**31**              Decode packet into a set of *combine* task descriptors using $T_\phi$, $G_\phi, O$

**32**              Write task descriptors to $tQ$

**33**              Notify Scheduler of decoded tasks

**34**           **end if**

**35**        **end**

**36**     **end while**

**37 end**

---

# G Implementation

Table 4: Implementation metrics of FlashMoE.

| Metric | Value |
|---|---|
| Total lines of code (CUDA/C++) | 6820 |
| Kernel stack frame size | 0 B |
| Spill stores (per thread) | 0 |
| Spill loads (per thread) | 0 |
| Shared memory usage (per block) | 46 KB |
| Registers per thread | 255 |
| Max active blocks per SM | 2 |
| Compilation time | 53 seconds |
| Binary size | 29 MB |

# H FP16 Memory Throughput

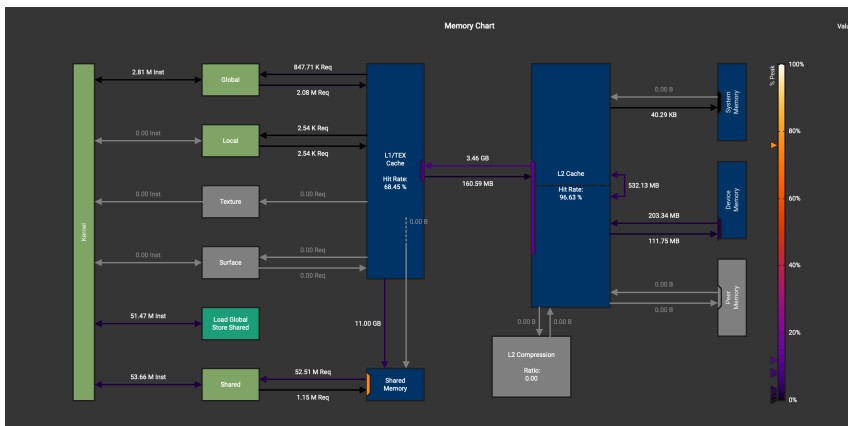

(a) Memory subsystem throughput for FP16

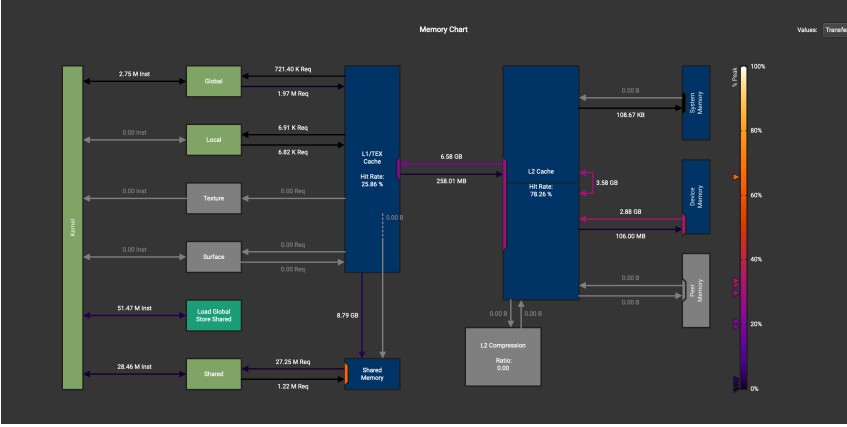

(b) Memory subsystem throughput for FP32

Figure 16: Here, we report the total A100 memory throughput for both FP16 (top) and FP32 (bottom) variants of FlashMoE. Notably, the FP16 implementation issues approximately $2\times$ more shared memory instructions compared to its FP32 counterpart under identical workloads. We attribute this inefficiency to suboptimal shared memory layouts in FlashMoE when operating on half-precision data. While this bottleneck is addressable through improved layout strategies, we leave its resolution to future work.

