# OpenReview forum: "FlashMoE: Fast Distributed MoE in a Single Kernel"
_NeurIPS.cc/2025/Conference — NeurIPS 2025 poster_

### Official Review · Reviewer_Kbkq · 2025-06-01

**Clarity:** 3
**Significance:** 2
**Originality:** 3
**Rating:** 3
**Confidence:** 4

**Summary:**

This paper introduces FlashDMoE, a novel distributed MoE operator that fuses computation and communication into a single persistent GPU kernel. The work is motivated by the growing inefficiencies in conventional MoE models where distributed experts incur high communication overheads due to frequent kernel launches and synchronous communication operations. The paper addresses the challenges by employing device-initiated asynchronous communication, fine-grained tile-level parallelism, and an actor-based scheduling model, resulting in significant speedups (up to 15×) over existing state-of-the-art MoE implementations.

**Questions:**

See weaknesses.

**Ethical Concerns:**

["NO or VERY MINOR ethics concerns only"]

**Limitations:**

Yes.

**Quality:**

3

**Strengths And Weaknesses:**

Strengths:

1. The approach introduces a single persistent GPU kernel that fuses all computation and communication tasks. It successfully eliminates repeated kernel launch overhead and reduces idle time on GPUs. It achieves higher GPU utilization and demonstrably faster execution compared to multiple short-lived kernel launches in existing systems.

2. The proposed method leverages device-initiated asynchronous communication to mitigate the straggler problem. It efficiently routes only active tokens to the GPUs hosting the relevant experts. It minimizes excessive data transfer by implementing payload-efficient communication.

3. The design employs fine-grained tile-level parallelism and an actor-based concurrency model. It effectively partitions the workload into small, independent tasks that can be dynamically scheduled across GPU warps. It optimizes the balance between computation and administrative tasks for better overall performance.

4. The experimental evaluations demonstrate strong performance improvements across multiple dimensions. It shows substantial speedups when scaling tokens, experts, and GPUs compared to existing MoE implementations. It substantiates the claims with detailed profiling and comparative analyses.


Weaknesses:


1. The current implementation targets FP32 precision exclusively. It does not explore or support lower-precision arithmetic such as FP16 or mixed precision, which are common in modern deep learning deployments. This limitation may impact its applicability and performance in production-scale environments where lower precision is preferred.

2. While optimizing the efficiency of MoE at the CUDA level is helpful and practical, from the system perspective, the contribution of kernel fusion is relatively weak. A simple kernel fusion is trivial and might be not enough for NeurIPS.

---

> ### Author Rebuttal · Authors · 2025-07-30
>
> We thank reviewer Kbkq for evaluating our paper and providing constructive feedback. Here, we aim to address the points raised regarding FP32 precision of FlashMoE and provide a more in-depth understanding of the novelty of FlashDMoE.
>
> ## W1: Precision
> We acknowledge the concern regarding FlashDMoE’s current use of FP32 precision. We note that our evaluation in the paper using FP32 represents a conservative estimate of FlashDMoE’s potential benefits. Despite operating in full precision, FlashDMoE substantially outperforms baseline systems that already utilize FP16 precision. Thus, transitioning FlashDMoE to FP16 precision will further improve these already significant performance gains.
>
> On another note, we acknowledge the use of fp32 as a limitation in our paper and we highlight that FlashDMoE already supports half precision in bf16 and fp16. However, our fp16 implementations do not yet achieve peak hardware utilization due to (1) suboptimal shared memory layouts and (2) insufficient vectorization in custom computational operators within the Processor. To overcome these limitations, we plan to integrate recent advancements from Device Extensions (DX) libraries [1]. These libraries build on top of the CUTLASS APIs and provide a specialized configuration layer with highly optimized parameters, significantly improving both hardware memory throughput and tensor-core utilization. We anticipate adopting [1] in FlashDMoE and, in fact, already have ongoing work to that effect. In summary, this transition would involve deprecating the CollectiveMainLoop defined in `mmaConfig.cuh` and line 80 of `gemm.cuh`, in favor of a bespoke GEMM main loop that accumulates across the K dimension, where each iteration dispatches to the highly-tuned, block-scoped operator of [1]. Importantly, [1]’s adherence to returning GEMM results as fragments stored in fast register memory seamlessly preserves FlashDMoE’s ability to execute post-GEMM fusion operations. We plan to make these changes available in our public codebase.
>
> ## W2: Novelty
> We acknowledge that _simple_ kernel fusion is a straightforward task. However, achieving _fast, distributed_ kernel fusion is a significant challenge, particularly for MoE models, which expert researchers characterize as "hard to train" [2]. FlashDMoE is the first to introduce a fully fused, high-performance, systematic distributed MoE operator; to the best of our knowledge, previous systems [3, 4] have only achieved partial fusion through ad-hoc methods.
>
> On one hand, efficient kernel fusion requires an in-depth understanding of the GPU memory hierarchy, its weak memory model, and extensive experience with its programming abstractions, particularly multidimensional thread-value layouts for coalesced access and the Matrix-Multiply-Accumulate (MMA) instructions provided by specialized libraries such as CUTLASS. On the other hand, implementing these advanced techniques within a distributed framework constitutes the core innovation of our work. Specifically, we present (1) an in-kernel actor-based operating system that enables high-throughput distributed tile parallelism and (2) a symmetric tensor layout that facilitates completely non-blocking and provably conflict-free memory access across distributed GPUs and tens of thousands of GPU threads.
>
> Thus, the novelty of our contributions, combined with their superior performance compared to baseline works [3, 5] from prestigious venues, positions FlashDMoE, we believe, as an excellent fit for NeurIPS.
>
> ## References
>
> [1] NVIDIA. cuBLASDx.
>
> [2] Mosaic Research. _Introducing DBRX: A New State-of-the-Art Open LLM_. 2024.
>
> [3] Zhang et al. _COMET_. MLSys ‘25, Outstanding Paper Honorable Mention.
>
> [4] DeepSeek-AI. _DeepEP_. 2025.
>
> [5] He et al. _FasterMoE_, PPopp’22.

---

> > ### Comment · Reviewer_Kbkq · 2025-08-01
> >
> > Thanks for the rebuttal. My concerns are not addressed. I'd like to keep my score.

---

> > > ### Author Response · Authors · 2025-08-06
> > >
> > > We sincerely appreciate your thoughtful feedback and respect your decision. We wish to clarify why we believe FlashDMoE’s contribution is significant:
> > >
> > > **On Precision:** We recognize your concern regarding FP32. Although our evaluation was presented in FP32, FlashDMoE already supports FP16. Our ongoing transition to an optimized FP16 implementation with improved tensor-core utilization (leveraging DX libraries/CUTLASS) will further amplify the already significant benefits of FlashdMoE. Our codebase is open-source, underscoring the practicality and potential of our approach.
> > >
> > > **On Novelty of Kernel Fusion:** We recognize that basic kernel fusion may appear trivial. However, FlashDMoE goes well beyond simple fusion by introducing an entirely new in-kernel distributed runtime: an actor-based scheduling model and conflict-free symmetric tensor layout designed specifically for distributed MoE systems. These innovations were not previously achieved in existing high-quality work (COMET, DeepEP, FasterMoE and others), which relied on partial fusion. Thus, FlashDMoE sets a new standard for systematic distributed MoE optimization.
> > >
> > > We appreciate your careful consideration and hope this additional context clarifies the significance of our contributions.

---

### Official Review · Reviewer_pAJq · 2025-06-22

**Clarity:** 4
**Significance:** 4
**Originality:** 3
**Rating:** 4
**Confidence:** 4

**Summary:**

This paper introduces FlashDMoE, a system designed to accelerate the execution of distributed Mixture-of-Experts (MoE) layers in large language models.

The work identifies two bottlenecks in existing MoE implementations: communication and kernel launch overheads.

To solve these problems, the authors propose FlashDMoE, an MoE operator architected as a single persistent GPU kernel. Instead of relying on the CPU to launch a sequence of kernels, FlashDMoE launches a monolithic kernel that remains active throughout the entire MoE operation, managing all internal tasks directly on the GPU to eliminate kernel launch overhead by design, reducing the number of required GPU operations from hundreds to one.

The core mechanisms of FlashDMoE are threefold:

1. Actor-Based Concurrency
2. Asynchronous, Device-Initiated Communication
3. Payload-Efficient Data Transfer

The authors present a comprehensive evaluation of multi-GPU servers equipped with NVIDIA H100s. The results demonstrate that FlashDMoE achieves speedups ranging from 1.2x to 15x over state-of-the-art baselines. Furthermore, the system achieves over 90% Streaming Multiprocessor (SM) utilization, showcasing its effectiveness at eliminating GPU idle time.

**Questions:**

Please see above.

**Ethical Concerns:**

["NO or VERY MINOR ethics concerns only"]

**Final Justification:**

I am keeping my score.

**Limitations:**

Yes.

**Paper Formatting Concerns:**

No.

**Quality:**

3

**Strengths And Weaknesses:**

Strength:

1. The idea is intuitive and addresses bottlenecks in traditional LLM kernels.
2. The experiments show good latency hiding and resource utilization results.

Weakness

1. While the idea of a monolithic kernel is new, the core techniques are not. The idea of integrating communication and computation tasks within fused GPU kernels and thread block specialization is well-studied and are in one of the baselines, Comet, that the authors compare with. Other than pushing the idea to a single kernel, which comes with its drawbacks of less flexibility, it is not clear what the additional challenges are.

2. It's not clear how the additional activation buffer would affect per-device batch size in a larger-scale scenario. If this extra memory requirement causes higher GPU requirements, then a more detailed analysis of the efficiency-memory tradeoff is needed.

3. FP32 is a suboptimal setup for both inference and training.

4. It's not clear how much additional effort is required to support more diverse expert routing algorithms.

Minor typos:

1. Extra comma in line 43.
2. Incomplete/fragmented sentence in line 138.
3. Duplicated sentence in line 205.

---

> ### Author Rebuttal · Authors · 2025-07-30
>
> We thank you for your constructive comments on our work. We provide details to address your concerns (weaknesses W1–W4):
>
> ## W1: Novelty of FlashDMoE
> We acknowledge that simple kernel fusion is a straightforward task. However, achieving fast, distributed kernel fusion is a significant challenge, particularly for MoE models, which expert researchers characterize as "hard to train" [1]. FlashDMoE is the first to introduce a fully fused, high-performance, systematic distributed MoE operator; to the best of our knowledge, previous systems [2, 3] have only achieved partial fusion through ad-hoc methods.
>
> On one hand, efficient kernel fusion requires an in-depth understanding of the GPU memory hierarchy, its weak memory model, and extensive experience with its programming abstractions, particularly multidimensional thread-value layouts for coalesced access and the Matrix-Multiply-Accumulate (MMA) instructions provided by specialized libraries such as CUTLASS. On the other hand, implementing these advanced techniques within a distributed framework constitutes the core innovation of our work. Specifically, we present (1) an in-kernel actor-based operating system that enables high-throughput distributed tile parallelism and (2) a symmetric tensor layout that facilitates completely non-blocking and provably conflict-free memory access across distributed GPUs and tens of thousands of GPU threads.
>
> ## W2: Memory Footprint
> We acknowledge the concern that FlashDMoE requires extra memory for the symmetric tensor (Section 3.2). We maintain that this overhead is modest in comparison with the existing memory requirements of MoE models. To show this empirically, we compute and analyze memory costs for popular large MoE models:
>
> | **Model**          | **Params** | **S** | **E** | **H** | **Intermediate** | **Symmetric Tensor (GB)** | **Memory for Model (GB)** | **Overhead Ratio (%)** |
> |--------------------|------------|-------|-------|-------|------------------|---------------------------|---------------------------|------------------------|
> | Moonlight-16B-A3B  | 16B        | 8K    | 64    | 2K    | 1.375K           | 0.25                      | 59                        | **0.49**               |
> | Grok-1             | 314B       | 8K    | 8     | 6K    | 32K              | 0.75                      | 1169                      | **0.15**               |
> | Snowflake-Arctic   | 479B       | 4K    | 128   | 7K    | 4.75K            | 1.75                      | 1784                      | **0.12**               |
> | Qwen3-235B-A22B    | 235B       | 40K   | 128   | 4K    | 1.5K             | 3                         | 875                       | **0.38**               |
> | Mixtral 8x7B       | 47B        | 32K   | 8     | 4K    | 14K              | 2                         | 175                       | **2.15**               |
> | DeepSeek-V3        | 685B       | 160K  | 256   | 7K    | 2K               | 1.5                       | 2551                      | **0.11**               |
>
>
> We calculate the model’s memory requirement by multiplying its parameter count with precision (fp32 here but this choice is inconsequential to the ratio which we embolden). Note that the symmetric tensor demands at most 2% extra memory in comparison to the inference memory requirements. This overhead would even be more amortized in training, where the memory demand for the model is much greater. We will be sure to add a more thorough empirical analysis elaborating on the memory and performance tradeoffs exhibited in FlashDMoE.
>
> ## W3: Precision
> We acknowledge the concern regarding FlashDMoE’s current use of FP32 precision. We note that our evaluation in the paper using FP32 represents a conservative estimate of FlashDMoE’s potential benefits. Despite operating in full precision, FlashDMoE substantially outperforms baseline systems that already utilize FP16 precision. Thus, transitioning FlashDMoE to FP16 precision will further improve these already significant performance gains.
>
> On another note, we acknowledge the use of fp32 as a limitation in our paper and we highlight that FlashDMoE already supports half precision in bf16 and fp16. However, our fp16 implementations do not yet achieve peak hardware utilization due to (1) suboptimal shared memory layouts and (2) insufficient vectorization in custom computational operators within the Processor. To overcome these limitations, we plan to integrate recent advancements from Device Extensions (DX) libraries [4]. These libraries build on top of the CUTLASS APIs and provide a specialized configuration layer with highly optimized parameters, significantly improving both hardware memory throughput and tensor-core utilization. We anticipate adopting [4] in FlashDMoE and, in fact, already have ongoing work to that effect. In summary, this transition would involve deprecating the CollectiveMainLoop defined in `mmaConfig.cuh` and line 80 of `gemm.cuh`, in favor of a bespoke GEMM main loop that accumulates across the K dimension, where each iteration dispatches to the highly-tuned, block-scoped operator of [4]. Importantly, [4]’s adherence to returning GEMM results as fragments stored in fast register memory seamlessly preserves FlashDMoE’s ability to execute post-GEMM fusion operations. We plan to make these changes available in our public codebase.
>
>
> ## W4: Diverse Expert Routing
> FlashDMoE already supports unbounded k for expert top-k selection and can accommodate up to 65,535 experts—significantly exceeding the architectural specifications of any state-of-the-art MoE model and the memory requirements of modern GPUs.
>
> ### Typos
> We appreciate you identifying the typos! We will fix them.
>
> ## References
>
> [1] Mosaic Research. _Introducing DBRX: A New State-of-the-Art Open LLM_. 2024.
>
> [2] Zhang et al. _COMET_. MLSys ‘25, Outstanding Paper Honorable Mention.
>
> [3] DeepSeek-AI. _DeepEP_. 2025.
>
> [4] NVIDIA. cuBLASDx.

---

### Official Review · Reviewer_KK8z · 2025-07-02

**Clarity:** 3
**Significance:** 3
**Originality:** 3
**Rating:** 4
**Confidence:** 2

**Summary:**

Although MoE models are efficient because they only activate a few experts at a time, spreading experts across many GPUs causes heavy communication and frequent GPU launches, especially due to ALLTOALL operations. To solve this, FlashDMoE combines all computing and communication steps into one long-running GPU kernel, removing the need for CPU control.

**Questions:**

Have you trained an MoE model to compare the stability, throughput, and loss of FlashDMoE and Megatron during actual training?

**Ethical Concerns:**

["NO or VERY MINOR ethics concerns only"]

**Final Justification:**

I appreciate the authors' response. It has addressed my concerns, so I decide to keep my original rating.

**Limitations:**

yes

**Paper Formatting Concerns:**

There is no format problem.

**Quality:**

2

**Strengths And Weaknesses:**

Strengths:
* It achieves significant speedups, with 1.2X-15X speedup over state-of-the-art MoE implementations in evaluations on multiple ML servers each with 8 NVIDIA H100 GPUs, and 3−10× improvement in forward pass latency on a single node with 8 NVlink-ed H100 GPUs compared to existing MoE systems.
* It eliminates repeated kernel launches, launching only one persistent kernel during a single forward pass, while baselines launch up to 550 short-lived kernels, thus reducing kernel launch overheads significantly.
* It mitigates the impact of communication stragglers through fine-grained, tile-level parallelism within the kernel, addressing the long-tail latency issue of synchronous ALLTOALL communication.
* It improves payload efficiency by introducing in-place padding and symmetric tensor layouts, transmitting only active tokens, and reducing unnecessary data transfers, which leads to better runtime performance.

Weaknesses:
* FlashDMoE is currently only evaluated on a single node with 8 GPUs connected via NVLink. Its performance on multi-node systems, especially over slower interconnects like InfiniBand, is not yet tested.
* To enable non-blocking communication, FlashDMoE over-provisions memory using a 4× symmetric tensor layout, which increases memory consumption compared to some other MoE implementations.
* The system is benchmarked on synthetic workloads and toy MoE layers, without integration into full-scale production models like LLaMA or GPT. Thus, the practical benefit of end-to-end training/inference remains to be validated.

---

> ### Author Rebuttal · Authors · 2025-07-30
>
> We thank you for your constructive feedback on our work. We provide details to address your concerns (weaknesses W1–W3).
>
> ## W1: Multi-node experiments with FlashDMoE
> We acknowledge that the main body of the paper did not present results of FlashDMoE on multi-node hardware. To address this concern, we successfully set up FlashDMoE on 4 GPU nodes, where each node contains 4 A100 80G GPUs (16 A100 GPUs in total), on a Supercomputer in the United States. Every GPU on a node has a dedicated NIC with 25 GB/s of bandwidth. There are 16 experts in this experiment. We provide preliminary latency results on this multi-node setup:
>
> | Tokens | FlashDMoE Latency (ms) |
> |--------|--------------|
> | 128    | 1.22         |
> | 256    | 1.26         |
> | 512    | 1.29         |
> | 1024   | 1.43         |
> | 2048   | 1.74         |
>
> We note that these results do not test all possible configurations in terms of number of tokens and experts. Due to limited time, these were the only configurations we managed to test. But, we hope they address some reviewer concerns regarding multi-node evaluation of FlashDMoE. We intend to run FlashDMoE across multiple nodes with a full set of configurations and compare with popular baselines in the near future. We will include these results in the camera-ready version of the paper (if accepted).
>
> ## W2: Memory footprint of the Symmetric Tensor
> We acknowledge the concern that FlashDMoE requires extra memory for the symmetric tensor (Section 3.2). We maintain that this overhead is modest in comparison with the existing memory requirements of MoE models. To show this empirically, we compute and analyze memory costs for popular large MoE models:
>
> | **Model**          | **Params** | **S** | **E** | **H** | **Intermediate** | **Symmetric Tensor (GB)** | **Memory for Model (GB)** | **Overhead Ratio (%)** |
> |--------------------|------------|-------|-------|-------|------------------|---------------------------|---------------------------|------------------------|
> | Moonlight-16B-A3B  | 16B        | 8K    | 64    | 2K    | 1.375K           | 0.25                      | 59                        | **0.49**               |
> | Grok-1             | 314B       | 8K    | 8     | 6K    | 32K              | 0.75                      | 1169                      | **0.15**               |
> | Snowflake-Arctic   | 479B       | 4K    | 128   | 7K    | 4.75K            | 1.75                      | 1784                      | **0.12**               |
> | Qwen3-235B-A22B    | 235B       | 40K   | 128   | 4K    | 1.5K             | 3                         | 875                       | **0.38**               |
> | Mixtral 8x7B       | 47B        | 32K   | 8     | 4K    | 14K              | 2                         | 175                       | **2.15**               |
> | DeepSeek-V3        | 685B       | 160K  | 256   | 7K    | 2K               | 1.5                       | 2551                      | **0.11**               |
>
> We calculate the model’s memory requirement by multiplying its parameter count with precision (fp32 here but this choice is inconsequential to the ratio which we embolden). Note that the symmetric tensor demands at most 2% extra memory in comparison to the inference memory requirements. This overhead would even be more amortized in training, where the memory demand for the model is much greater. We will be sure to add a more thorough empirical analysis elaborating on the memory and performance tradeoffs exhibited in FlashDMoE.
>
> ## W3: Integration
> We acknowledge that our evaluation benchmarks MoE layers rather than full-scale production models like LLaMA or GPT. While our experiments rigorously quantify fundamental performance improvements from having a fully-fused distributed MoE implementation, validating these benefits through integration into production models remains our future work. We are actively investigating this line of work and are eagerly look forward to showcasing the performance of FlashDMoE in realistic training and inference scenarios.

---

> > ### Author Response · Authors · 2025-08-05
> >
> > We hope that we have addressed your concerns in the rebuttal, and we are happy to answer any questions you may have.

---

### Official Review · Reviewer_MD3K · 2025-07-03

**Clarity:** 2
**Significance:** 3
**Originality:** 3
**Rating:** 4
**Confidence:** 3

**Summary:**

This paper proposed FlashDMoE, a fully fused MoE operator that runs as a single GPU kernel. It integrates all compute and communication tasks directly on the GPU, bypassing the need for multiple kernel launches and improving GPU utilization. The innovative of this paper include 1. replacing hundreds of short-lived GPU kernels with one long-lived kernel; 2. breaking work into tiles processed in parallel across GPU threads; 3. communicating only with GPUs hosting selected experts. The experiment result shows it achieves 1.2× to 15× faster performance compared to SOTA method like DeepEP, Megatron-LM, FasterMoE, and Comet.

**Questions:**

1. How difficult would it be to adapt FlashDMoE to support other MoE routing strategies, such as top-k routing with dynamic capacity?
2. How easily can FlashDMoE be integrated into popular frameworks like Transformers or PyTorch?
3. Do you plan to open-source the implementation? If so, will it include utilities for configuring different model shape and architecture?

**Ethical Concerns:**

["NO or VERY MINOR ethics concerns only"]

**Final Justification:**

I would give this paper a borderline accept because it is a complete work with easy-use design, sufficient experiments, and well-written manuscripts.

**Limitations:**

Please refer to weakness and questions.

**Paper Formatting Concerns:**

The font size in the figures is slightly little.

**Quality:**

2

**Strengths And Weaknesses:**

Strengths:
1. Novel Single-Kernel Design: FlashDMoE introduces a unified, persistent GPU kernel that eliminates host-side kernel launch overhead.
2. Deep Systems Innovation: The paper presents a well-engineered actor-based GPU execution model with tile-level parallelism and efficient memory management.
3. Substantial Performance Gains: The approach achieves 1.2× to 15× speedup over existing MoE systems and significantly improves GPU utilization.

Weaknesses:
1. Lack of Open Source and Flexibility: The code is not open-sourced, and adapting the implementation to new MoE architectures or shapes may be complex.
2. Limited Evaluation Scope: All experiments are conducted on a single-node, 8-GPU system, with no multi-node benchmarks.
3. Missing Baseline Comparisons: The paper does not compare against other recent expert-parallel fusion methods, such as [Optimizing Distributed ML Communication with Fused Computation-Collective Operations](https://arxiv.org/html/2305.06942v2).

---

> ### Author Rebuttal · Authors · 2025-07-30
>
> We thank reviewer MD3K for their constructive feedback. Below, we address the concerns raised about open-source nature and flexibility of FlashDMoE, the limitations of our evaluation framework, and the absence of baseline comparisons. We hope this response highlights the robustness of our approach and its potential for future enhancements.
>
> ## W1: Open Source Code
>
> We echo the raised concern about open-source software, hence why we included our open-source, workable code in the supplementary material of this submission. We currently have a well-maintained, public GitHub repository (under a different name to preserve anonymity) with over 30 stars and a rapidly growing user base. To preserve the integrity of the review process and maintain anonymity, we refrained from including links to this repository in the manuscript but reiterate that we provide the code in supplemental materials.
>
> ## W1: Flexibility of FlashDMoE
>
> FlashDMoE can already accommodate arbitrary dimensions for sequence length, token embedding dimensions, and FNN intermediate size. The included JSON file (`aristos_config.json`) in the included FlashDMoE codebase allows users to specify these parameters. Additionally, we support unbounded k for expert top-k selection, offer four different precisions (fp16, bf16, tf32, and fp32), and can accommodate up to 65,535 experts—significantly exceeding the architectural specifications of any state-of-the-art MoE model and the memory requirements of modern GPUs.
>
> We note that in its current implementation, FlashDMoE does not support alternative MoE architectures like the shared expert approach used in DeepSeekMoE [1] or the Pyramid-Residual MoE technique introduced in DeepSpeedMoE [2]. Yet, we optimistically highlight that the key ingredients for unlocking these architectural shifts, and future ones, already exist in the design of FlashDMoE: tile-parallel implementation and task abstraction for computation. Specifically, (1) underlying these MoE variants are computational operators that are describable in our task framework (Section 3.1), and (2) for expert parallelism, the symmetric tensor (Section 3.2) will still support scalable inter- or intra-GPU communication among all three actors, whose algorithms remain structurally unchanged across these architectural variations.
>
> ## W2: Limited Evaluation on multi-node setups
>
> We acknowledge that the main body of the paper did not present results of FlashDMoE on multi-node hardware. To address this concern, we successfully set up FlashDMoE on 4 GPU nodes, where each node contains 4 A100 80G GPUs (16 A100 GPUs in total), on a Supercomputer in the United States. Every GPU on a node has a dedicated NIC with 25 GB/s of bandwidth. There are 16 experts in this experiment. We provide preliminary results of the FlashdMoE operator latency on this multi-node setup:
>
> | Tokens | FlashdMoE Latency (ms) |
> |--------|--------------|
> | 128    | 1.22         |
> | 256    | 1.26         |
> | 512    | 1.29         |
> | 1024   | 1.43         |
> | 2048   | 1.74         |
>
> We note that these results do not test all possible configurations in terms of number of tokens and experts. Due to limited time, these were the only configurations we managed to test. But, we hope they address some reviewer concerns regarding multi-node evaluation of FlashDMoE. We intend to run FlashDMoE across multiple nodes with a full set of configurations and compare with cutting-edge baselines in the near future. We will include these results in the camera-ready version of the paper (if accepted).
>
> ## W3: Missing Baseline Comparison
>
> Thank you for pointing out this related work; we highlight that we discussed and cited this SC'24 paper in the related works section (Section C) of our manuscript. To the best of our knowledge, we could not find an open-source implementation provided by the authors. We will make an effort to replicate their fused GEMM+All-to-All kernel for comparison with FlashDMoE.
>
> ## Q1: MoE Architectural variants
>
> Please see **W1: Flexibility of FlashDMoE**. Furthermore, we will be sure to include a more thorough discussion about how FlashDMoE can adapt to these architectural variants in the camera-ready version of the paper (if accepted).
>
>
> ## Q2: Integration to PyTorch
>
> Thanks for asking! FlashDMoE exposes a minimalistic forward API, line 73 of `moe.cuh`, which we anticipate upstreaming as a Python API that integrates with frameworks like PyTorch, HuggingFace Transformers or Megtron-LM. We hope to complete this Python release before the camera-ready deadline.
>
> ## Q3: Open Sourcing FlashDMoE
>
> Kindly see **W1: Open Source Code**
>
> ## References
>
> [1] DeepSeek-AI. DeepSeek-V2, arXiv, 2024.
>
> [2] Rajbhandari et al. DeepSpeed-MoE, ICML ‘22.

---

> > ### Comment · Reviewer_MD3K · 2025-08-04
> > **Rebuttal reviewer response**
> >
> > The authors have provided satisfactory clarifications and demonstrations, and the research manuscript presents significant technical merit. I recommend its acceptance.

---

### Official Review · Reviewer_ST1R · 2025-07-03

**Clarity:** 3
**Significance:** 3
**Originality:** 3
**Rating:** 5
**Confidence:** 3

**Summary:**

This paper introduces FlashDMoE, a novel system for executing the Mixture-of-Experts (MoE) layer in distributed deep learning environments. The authors identify two critical performance bottlenecks in existing MoE implementations: (1) high communication overhead and straggler sensitivity from synchronous "ALLTOALL" collectives, and (2) significant latency from the host CPU launching hundreds of individual GPU kernels.

To address these issues, the authors propose a paradigm shift: fusing the entire MoE operator—including gating, expert computation, and inter-GPU communication—into a single, persistent GPU kernel. This design eliminates kernel launch overhead and enables fine-grained, asynchronous execution. The core technical contributions are:

1. An actor-based concurrency model within the kernel, where GPU warps are specialized into roles (Processor, Scheduler, Subscriber) to manage tasks.
2. A device-initiated asynchronous communication protocol built on a novel symmetric tensor layout. This layout overprovisions memory to create a partitioned global address space (PGAS), enabling non-blocking data transfers without synchronization penalties.

Payload-efficient communication via "in-place padding", which avoids transmitting zero-padded tokens across the network.

The authors evaluate FlashDMoE on an 8x H100 GPU server, demonstrating significant forward-pass latency speedups of 1.2x to 15x over state-of-the-art systems like DeepEP, Comet, and Megatron-LM.

**Questions:**

Q1. How sensitive is the performance of the symmetric tensor layout to the number of communication rounds ($\tau$)? The paper states $2\cdot\tau$=4 for MoE, but how would the design and its memory overhead adapt to more complex computational graphs with more communication steps?

Q2. The actor model dedicates one warp to the Scheduler role. Have you observed this scheduler becoming a bottleneck in any configuration, particularly with a very high number of experts or small tile sizes, leading to a high rate of task dispatching?

Q3. The evaluation focuses exclusively on the forward pass for inference. Have you considered the implications of this design for the backward pass in a training scenario? Or, you can modify the architecture in a training-free manner? How would the actor model and communication scheme handle the additional complexities of gradient computation and accumulation?

**Ethical Concerns:**

["NO or VERY MINOR ethics concerns only"]

**Final Justification:**

I will keep my score.

**Quality:**

3

**Strengths And Weaknesses:**

S1. The concept of a fully fused, single-kernel MoE operator is a radical and elegant departure from conventional approaches. It fundamentally rethinks the execution model to eliminate well-known bottlenecks at their source, rather than merely optimizing around them. If this approach proves generalizable, it could influence the design of other complex communication-intensive operators in deep learning.

S2. The experimental results are compelling. Achieving speedups of up to 15x and maintaining over 90% SM utilization is a testament to the effectiveness of the design. The system's ability to maintain near-constant latency when scaling the number of tokens (Figure 10a) is particularly noteworthy and highlights the success of the computation-communication overlap.

S3. The symmetric tensor layout is a intuitive and insightful solution to the challenge of asynchronous, one-sided communication. Trading a predictable amount of memory for the complete removal of synchronization barriers is a powerful and well-justified engineering decision. The actor model provides a clean abstraction for managing the complex interactions within the fused kernel.

S4. The authors conduct a comprehensive evaluation against multiple strong baselines. The experiments are well-designed to isolate performance across different scaling dimensions (tokens, experts, GPUs). The inclusion of micro-benchmarks that analyze GPU utilization (Fig 12a) and payload efficiency (Fig 12b) provides clear evidence for why the system is faster, strengthening the paper's claims.

W1.The primary concern is the 4x memory overhead required for the token buffer to enable the symmetric tensor layout. The authors describe this as "modest," but for models with very large sequence lengths or on memory-constrained hardware, this could be a prohibitive cost. The paper would benefit from a more detailed analysis of this trade-off, perhaps identifying the point at which this memory cost outweighs the performance benefits.

W2. The claims of the paper are based entirely on a single-node, multi-GPU setup with high-bandwidth NVLink interconnect. The most challenging ALLTOALL bottlenecks and straggler effects often manifest in multi-node clusters connected over lower-bandwidth networks like InfiniBand. Without multi-node results, the paper's claims about scalability and straggler mitigation remain partially unsubstantiated. This is the most significant gap in the current evaluation.

W3. The implementation is restricted to FP32 precision. Modern LLM inference and training workflows heavily rely on lower-precision formats (FP16, BF16, FP8) for performance and memory efficiency. This limitation currently makes FlashDMoE impractical for many state-of-the-art models.

W4. The system is a highly specialized, low-level implementation (6,800+ lines of CUDA/C++). This presents a potential barrier to adoption and maintenance compared to solutions built on standard, well-documented libraries like NCCL. While not a flaw in the research, it's a practical consideration worth acknowledging.

---

> ### Author Rebuttal · Authors · 2025-07-30
>
> We thank you for your constructive comments and glowing commendations of our work. Below, we address your concerns (weaknesses W1–W4) and answer your thoughtful questions (Q1–Q3):
>
> ## W1: Memory overhead for the Symmetric Tensor
>
> We acknowledge the concern that FlashDMoE requires extra memory for the symmetric tensor (Section 3.2). We maintain that this overhead is modest in comparison with the existing memory requirements of MoE models. To show this empirically, we compute and analyze memory costs for popular large MoE models:
>
> | **Model**          | **Params** | **S** | **E** | **H** | **Intermediate** | **Symmetric Tensor (GB)** | **Memory for Model (GB)** | **Overhead Ratio (%)** |
> |--------------------|------------|-------|-------|-------|------------------|---------------------------|---------------------------|------------------------|
> | Moonlight-16B-A3B  | 16B        | 8K    | 64    | 2K    | 1.375K           | 0.25                      | 59                        | **0.49**               |
> | Grok-1             | 314B       | 8K    | 8     | 6K    | 32K              | 0.75                      | 1169                      | **0.15**               |
> | Snowflake-Arctic   | 479B       | 4K    | 128   | 7K    | 4.75K            | 1.75                      | 1784                      | **0.12**               |
> | Qwen3-235B-A22B    | 235B       | 40K   | 128   | 4K    | 1.5K             | 3                         | 875                       | **0.38**               |
> | Mixtral 8x7B       | 47B        | 32K   | 8     | 4K    | 14K              | 2                         | 175                       | **2.15**               |
> | DeepSeek-V3        | 685B       | 160K  | 256   | 7K    | 2K               | 1.5                       | 2551                      | **0.11**               |
>
>
> We calculate the model’s memory requirement by multiplying its parameter count with precision (fp32 here but this choice is inconsequential to the ratio which we embolden). Note that the symmetric tensor demands **at most 2% additional memory** in comparison to the inference memory requirements. This overhead would even be more amortized in training, where the memory demand for the model is much greater. We will be sure to add a more thorough empirical analysis elaborating on the memory and performance tradeoffs exhibited in FlashDMoE.
>
> ## W2: Multi-node results with FlashDMoE
>
> We acknowledge that the main body of the paper did not present results of FlashDMoE on multi-node hardware. To address this concern, we successfully set up FlashDMoE on 4 GPU nodes, where each node contains 4 A100 80G GPUs (16 A100 GPUs in total), on a Supercomputer in the United States. Every GPU on a node has a dedicated NIC with 25 GB/s of bandwidth. There are 16 experts in this experiment. We provide preliminary results of the FlashDMoE operator latency on this multi-node setup:
>
> | Tokens | FlashDMoE Latency (ms) |
> |--------|--------------|
> | 128    | 1.22         |
> | 256    | 1.26         |
> | 512    | 1.29         |
> | 1024   | 1.43         |
> | 2048   | 1.74         |
>
> We note that these results do not test all possible configurations in terms of number of tokens and experts. Due to limited time, these were the only configurations we managed to test. But, we hope they address some reviewer concerns regarding multi-node evaluation of FlashDMoE. We intend to run FlashDMoE across multiple nodes with a full set of configurations and compare with popular baselines in the near future. We will include these results in the camera-ready version of the paper (if accepted).
>
> ## W3: Precision
> We acknowledge the concern regarding FlashDMoE’s current use of FP32 precision. We note that our evaluation in the paper using FP32 represents a conservative estimate of FlashDMoE’s potential benefits. Despite operating in full precision, FlashDMoE substantially outperforms baseline systems that already utilize FP16 precision. Thus, transitioning FlashDMoE to FP16 precision will further improve these already significant performance gains.
>
> On another note, we acknowledge the use of fp32 as a limitation in our paper and we highlight that FlashDMoE already supports half precision in bf16 and fp16. However, our fp16 implementations do not yet achieve peak hardware utilization due to (1) suboptimal shared memory layouts and (2) insufficient vectorization in custom computational operators within the Processor. To overcome these limitations, we plan to integrate recent advancements from Device Extensions (DX) libraries [1]. These libraries build on top of the CUTLASS APIs and provide a specialized configuration layer with highly optimized parameters, significantly improving both hardware memory throughput and tensor-core utilization. We anticipate adopting [1] in FlashDMoE and, in fact, already have ongoing work to that effect. In summary, this transition would involve deprecating the CollectiveMainLoop defined in `mmaConfig.cuh` and line 80 of `gemm.cuh`, in favor of a bespoke GEMM main loop that accumulates across the K dimension, where each iteration dispatches to the highly-tuned, block-scoped operator of [1]. Importantly, [1]’s adherence to returning GEMM results as fragments stored in fast register memory seamlessly preserves FlashDMoE’s ability to execute post-GEMM fusion operations. We plan to make these changes available in our public codebase.
>
> ## W4: Maintainability of FlashDMoE
> FlashDMoE's specialized CUDA/C++ implementation may pose challenges for maintenance, but not necessarily for _adoption_ as long as we employ ergonomic user abstractions. An apt case in point is NCCL, whose implementation details are quite intricate but exist behind well-defined and documented APIs that have effectively facilitated its adoption in deep learning frameworks like PyTorch. We envision FlashDMoE following the same trajectory as we already have designed ergonomic, user-friendly APIs that abstract the complexity of the fused kernel. Moreover, our low-level approach was critical for achieving significant performance gains over the state-of-the-art. We will make substantial efforts to highlight these points in the camera-ready version of the paper (if accepted).
>
> ## Q1: Performance sensitivity to increasing communication rounds
> Thank you for the question. In principle, our temporal buffering technique is not a functional requirement for our operator; less temporal slots $t$ necessitate locks or busy-waiting to ensure conflict-free access. Thus, we conceptualize $t$ as a tunable parameter that the user can modulate to navigate this tradeoff optimally in the general case, including for computational graphs where $t > 4$.
>
> ## Q2: Scheduler bottleneck
> Excellent question! The cost of dispatching 32 (32-way parallelism in a warp) tasks is equivalent to the latency of a global memory atomic operation, which is in the order of at least triple-digit nanoseconds. Even at tens of thousands of tasks, such latency only creeps to the double-digit microsecond regime, which is about the latency of computing a single 128x64 fp32 output tile. So, the bottleneck should not be task dispatching itself, but (1) how many processors are available at dispatch-time and (2) how quickly processors become available. Note that a processor's queue-depth is at most 1; it receives and processes only one task at a time and does not buffer. Thus, the more likely bottlenecking scenario is that the scheduler may have available tasks but no ready processors to dispatch to, as they are busy with computation or tile communication.
>
> ## Q3: Extensibility to training
> FlashDMoE is generalizable to training with modifications only needing to occur for (1) the Subscriber to account for encoding tasks required for the additional gradient GEMMs, and (2) additional task types to capture these functions. Communication would remain as-is, using the same symmetric layout and initiated eagerly by thread blocks when tile computation completes.
>
> ## References
>
> [1] NVIDIA. cuBLASDx.

---

### Note · Authors · 2025-08-12

We thank the reviewers for their thoughtful feedback. We are encouraged that we addressed your concerns during the rebuttal phase. Our final note summarizes key reservations brought up in the reviews and our responses to them:

1. **Memory overhead:** The overhead of FlashDMoE’s symmetric tensor is modest relative to the existing memory requirements of large MoE models. Empirically, our analysis of popular MoE architectures shows that the symmetric tensor introduces at most 2% additional memory compared to inference-time usage, and this cost is even further amortized during training, where total model memory demand is substantially higher.
2. **FP32 vs. lower precision implementation:** Our FP32 evaluation provides a conservative estimate of FlashDMoE’s benefits: even in full precision, it substantially outperforms FP16 baselines. FlashDMoE already supports half precision in BF16 and FP16. To achieve peak performance in lower precisions, we plan to integrate recent advancements from Device Extensions (DX) libraries into our implementation.
3. **Open-source code:** We included our open-source, workable code in the supplementary material of this submission. We maintain a public GitHub repository (under a different name to preserve anonymity) with 30+ stars and a rapidly growing user base.
4. **Multi-node experiments:** We successfully set up FlashDMoE on 4 GPU servers, where each node contains 4 A100 80G GPUs (16 A100 GPUs in total), on a supercomputer in the United States. Every GPU on a node has a dedicated NIC with 25 GB/s of bandwidth. There are 16 experts in this experiment. These experiments show that FlashDMoE exhibits strong performance even in multi-node settings.
5. **Novelty:** FlashDMoE sets itself apart from other research by developing the first fully-fused distributed MoE kernel. Our key innovation is bringing these techniques into a distributed setting through (1) an in-kernel actor-based OS enabling high-throughput distributed tile parallelism, and (2) a symmetric tensor layout that ensures fully non-blocking, conflict-free memory access across GPUs and tens of thousands of threads.

---

### Decision · Program_Chairs · 2025-09-17

**Decision:**

Accept (poster)

**Comment:**

**1. Scientific Claims and Findings**

This paper presents FlashDMoE, a novel system for accelerating distributed Mixture-of-Experts (MoE) layers by addressing two critical bottlenecks in existing implementations: high communication overhead from synchronous "ALLTOALL" operations and latency from frequent GPU kernel launches. The core claim is that fusing the entire MoE workflow—including gating, expert computation, and inter-GPU communication—into a single, persistent GPU kernel eliminates these bottlenecks. Key technical innovations include an actor-based concurrency model (specializing GPU warps into roles like Processor and Scheduler), a symmetric tensor layout enabling asynchronous, device-initiated communication, and "in-place padding" for payload-efficient data transfer. Evaluations on an 8x H100 GPU server demonstrate 1.2x to 15x forward-pass latency speedups over baselines (e.g., DeepEP, Megatron-LM) with >90% GPU utilization.

**2. Major Strengths**

The single-kernel design rethinks MoE execution, eliminating kernel launch overhead and enabling fine-grained computation-communication overlap—an elegant departure from conventional multi-kernel approaches. It reports significant speedups (up to 15x) and high GPU utilization (>90%) across diverse scenarios (scaling tokens, experts, GPUs). The symmetric tensor layout and asynchronous, device-initiated communication effectively mitigate straggler effects and reduce synchronization penalties, with "in-place padding" minimizing unnecessary data transfer. Comprehensive benchmarks against strong baselines, including micro-analyses of GPU utilization and payload efficiency, strengthen the credibility of performance claims.

**3. Major Weaknesses**

The 4x memory cost of the symmetric tensor layout may be prohibitive for large sequences or memory-constrained hardware, with insufficient analysis of this trade-off. Evaluations are restricted to single-node, 8x H100 setups with NVLink; multi-node performance over limited-bandwidth networks remains unproven, leaving scalability claims partially unsubstantiated. The restriction to data precision of FP32 (vs. low-precision formats like FP16/BF16) limits applicability to modern LLM training/inference workflows.

**4. Main reasons of accept**

The paper is recommended to be accepted due to its novel single-kernel design for MoE layers that effectively eliminates critical bottlenecks, delivers significant performance speedups, and introduces innovative communication and concurrency solutions supported by rigorous evaluations.

**5. Summary of discussions**

Most of the reviewers are satisfactory with the rebuttal. Reviewer Kbkq who gave rating of 3 had concerns about the precision and novelty of the work, and was not convinced by the rebuttal. By considering all the review comments and discussions, I am more inclined to recommend acceptance.